# Crystal structure generation with autoregressive large language modeling

Luis M. Antunes [1] ✉, Keith T. Butler [2] & Ricardo Grau-Crespo [1] ✉

The generation of plausible crystal structures is often the first step in predicting the structure and properties of a material from its chemical composition. However, most current methods for crystal structure prediction are computationally expensive, slowing the pace of innovation. Seeding structure prediction algorithms with quality generated candidates can overcome a major bottleneck. Here, we introduce CrystaLLM, a methodology for the versatile generation of crystal structures, based on the autoregressive large language modeling (LLM) of the Crystallographic Information File (CIF) format. Trained on millions of CIF files, CrystaLLM focuses on modeling crystal structures through text. CrystaLLM can produce plausible crystal structures for a wide range of inorganic compounds unseen in training, as demonstrated by ab initio simulations. Our approach challenges conventional representations of crystals, and demonstrates the potential of LLMs for learning effective models of crystal chemistry, which will lead to accelerated discovery and innovation in materials science.

The in silico search for new materials often involves the exploration of a space of compositions in a chemical system, and the investigation of various predicted structural phases in that space (see refs. [1],[2], and [3] for examples). To elucidate the structures of unknown materials, a Crystal Structure Prediction (CSP) approach is often employed, which attempts to derive the ground state crystal structure for a given chemical composition under specific physical conditions[4]. CSP approaches are relatively computationally expensive, typically involving ab initio techniques[5]. They often begin with the generation of candidate structures. Examples are the AIRSS[6,7] and USPEX[8] approaches. Initializing the search space with sensible structures increases the likelihood of success, and decreases the amount of computation required. It is therefore expected that effective crystal structure generation tools would help accelerate the prediction of structures using CSP methods.

Increasingly, techniques from machine learning (ML) and data science are being used to solve problems in materials science[9–11]. In particular, generative modeling approaches based on autoencoder architectures and generative adversarial networks[12] have been used to generate crystal structures[13–17]. Indeed, generative modeling has become commonplace, an outcome catalyzed by astounding advancements in the computational generation of images, audio and

natural language over the last several years[18]. The Large Language Model (LLM) approach, backed by the Transformer architecture[19], is behind state-of-the-art performance on natural language processing tasks. This approach begins with a generative pre-training step, which is autoregressive in nature, involving the unsupervised task of predicting the next token given a sequence of preceding tokens[20]. When such models are scaled to billions of parameters, their effectiveness becomes quite remarkable, as tools such as ChatGPT[21] demonstrate.

LLMs have recently been used in the context of materials science[22–28]. These attempts have been focused on using existing and publicly accessible LLMs, training, and tuning LLMs for natural language generation tasks involving chemical subject matter, or training LLMs on a corpus of expanded chemical compositions for the purposes of generating unseen compositions. However, the potential of training LLMs on textual representations of crystal structures has not been considered. A sole exception is a recent pre-print by Flam-Shepherd and Aspuru-Guzik, where the idea of generating the structures of molecules, materials, and protein binding sites with LLMs has been preliminarily explored[29].

Here, we report an LLM specifically designed for crystal generation. This model is distinctively trained on textual representations of

[1]Department of Chemistry, University of Reading, Whiteknights, Reading, UK. [2]Department of Chemistry, University College London, London, UK. ✉ e-mail: l.m.antunes@pgr.reading.ac.uk; r.grau-crespo@reading.ac.uk

inorganic crystal structures, specifically in the Crystallographic Information File (CIF) format[30], instead of relying solely on natural language corpora, or chemical compositions alone. The motivation for this approach originates from two conjectures: The first states that a sequence of symbols (i.e., tokens) is an appropriate representation modality for many predictive tasks, including those involving chemical structure. The idea of representing any domain with a sequence of tokens may at first seem counter-intuitive. However, consider that even images can be represented this way, and be subject to the autoregressive language modeling of pixels[31]. This challenges the notion that domain-specific representations, such as graphs for chemical structure[32], are necessary for superior performance. The second conjecture states that LLMs learn more than simply surface statistics and the conditional probability distribution of tokens. Indeed, autoregressive pre-training involving next-token prediction may result in learning an effective world model: an internalized causal model of the processes generating the target phenomena. A model which simply learns spurious correlations in the data is less desirable, as it may have greater difficulty in generalizing beyond the training distribution. Recent studies have demonstrated that LLMs trained on sequences of board game play (e.g., chess and Othello) do indeed track the state of the board, and probes of the internal activations of the model reveal the existence of representations of various abstract concepts specific to the domain[33,34]. We therefore asked whether a model trained to predict the 3-dimensional coordinates of atoms, digit-by-digit, could learn the chemistry implicit in crystal structures, and generate unseen structures, borrowing from its model of the world of atoms.

As such, we herein describe the CrystaLLM model, a tool for crystal structure generation trained on an extensive corpus of CIF files representing the structures of millions of inorganic solid-state materials. Unlike small molecule organic compounds, the generative modeling of inorganic crystals presents unique challenges: the structures are complex and periodic, are not readily described by simple graphs, are imbued with different forms of symmetry, and can be constructed from more than 100 different elements. Even so, the model is capable of reliably generating correct CIF syntax and physically plausible crystal structures for many classes of inorganic compounds. Moreover, we demonstrate how sampling from the model can be improved using the Monte Carlo Tree Search (MCTS) algorithm[35,36] together with a pre-trained graph-based neural network predictor of formation energy.

## Results
CrystaLLM is a Transformer-based, decoder-only language model of the CIF file format, trained autoregressively on a corpus of millions of CIF files (Fig. 1a). Rather than training on structural representations derived from the CIF files, the model is directly trained on the standardized and tokenized text contents of the CIF files. During training, the model is given a sequence of tokens from the corpus of CIF files, and is tasked with predicting the tokens which follow each of the given tokens. Once the model is trained, it can be used to generate new CIF files, conditioned on some starting sequence of tokens. Generating a CIF file involves repeatedly sampling tokens from the model, conditioning on the accumulated generated content, until a terminating condition is reached (Fig. 1b).

To assess the ability of the model to generate structures, a test set of ~10,000 randomly chosen CIF files is withheld from a training set of ~2.2 million CIF files, and the model is tasked with generating CIF files beginning from prompts constructed from the test set. Moreover, we assemble what we call a challenge set, which consists of 70 structures, 58 of which were obtained from the recent literature, and were not in the training set. The remaining 12 structures are from the training set, and are included as representatives of different structural classes. They serve to assess the model's ability to recover what it has seen in training, and as a means of comparing the model's generations of seen and unseen structures. (Supplementary Table 1 contains the full list of the challenge set compounds, and their sources.) The permutative nature of the dataset, with many structures having been derived by substituting atoms into pre-defined templates, results in a test set with the potential for some structures to closely resemble those of the training set. The challenge set provides a source of structures that are guaranteed to have been produced through a different process. Moreover, the challenge set constitutes a manageable set of compounds that reflects a variety of solid-state structural classes, allowing for a fine-grained picture of the model's capabilities. The test set, on the other hand, is better suited for a bulk assessment, and originates from the same distribution as the training set.

The following terminology is used in the remainder of this article: A formula, reduced formula, or reduced composition, refers to the empirical formula, or formula unit, which is the simplest, whole-number ratio of atoms in the compound. An example of a formula is $Ba_2MnCr$. A cell composition is a chemical formula referring to the total number of atoms of each type in the unit cell of a crystal. It represents the chemical formula of the compound as it would appear in the crystallographic unit cell, which might contain $Z$ formula units. An example of a cell composition is $Ba_6Mn_3Cr_3$, with a $Z$ of 3.

## Training and learned representations
Training consists of iteratively sampling sequences of tokens, of fixed length, and adjusting the model's parameters so that it becomes progressively better at predicting which token should follow a preceding sequence. (See the "Methods", and Supplementary Note 2, for more information on the model architecture and training.) Since it has been observed that LLM performance improves as the number of model parameters is increased[37], we train a small model, consisting of 25 million parameters, and a large model, consisting of 200 million parameters.

To monitor the progress of training, we withhold a validation set that constitutes 10% of the set held-out for training. Over the course of training, the model continues to improve in terms of its total cross-entropy loss on the validation set, even after 90,000 iterations (see Supplementary Fig. 2). We note, however, that improvements appear to become smaller with more training time.

As a consequence of the model's architecture, each token in a processed sequence is mapped to a distinct learned vector representation using an embedding table, whose parameters are adjusted during training. The result is that, through autoregressive training, distributed representations are learned for each symbol in the vocabulary. The vocabulary consists of symbols for atoms, space groups, and numeric digits. (See Supplementary Note 1 for a detailed description of the vocabulary and the tokenization procedure.) The training process appears to result in sensible representations of these various symbols. Plots of dimensionally-reduced atom and space group vectors demonstrate a logical structure, where similar entities cluster together, indicating that intrinsic properties and relationships are captured. (See Supplementary Fig. 3 for plots of the learned atom vectors, and Supplementary Fig. 4 for a plot of the learned space group vectors.) Moreover, examination of the learned numeric digit vectors reveals that numerical relationships are captured in the representations, as measurements of cosine and Euclidean distances between the learned digit vectors demonstrate a logical spatial relationship. (See Supplementary Fig. 5.) While not explored further in this work, we note that distributed representations of chemical entities, such as atoms, are useful for the prediction of materials properties[38,39].

## Generalizing to unseen structures
To evaluate the ability of the model to generate an unseen structure, the model is prompted with the structure's cell composition, and allowed to generate up to 3000 tokens. The prompt includes the first line of the CIF file, which consists of the data block header, containing

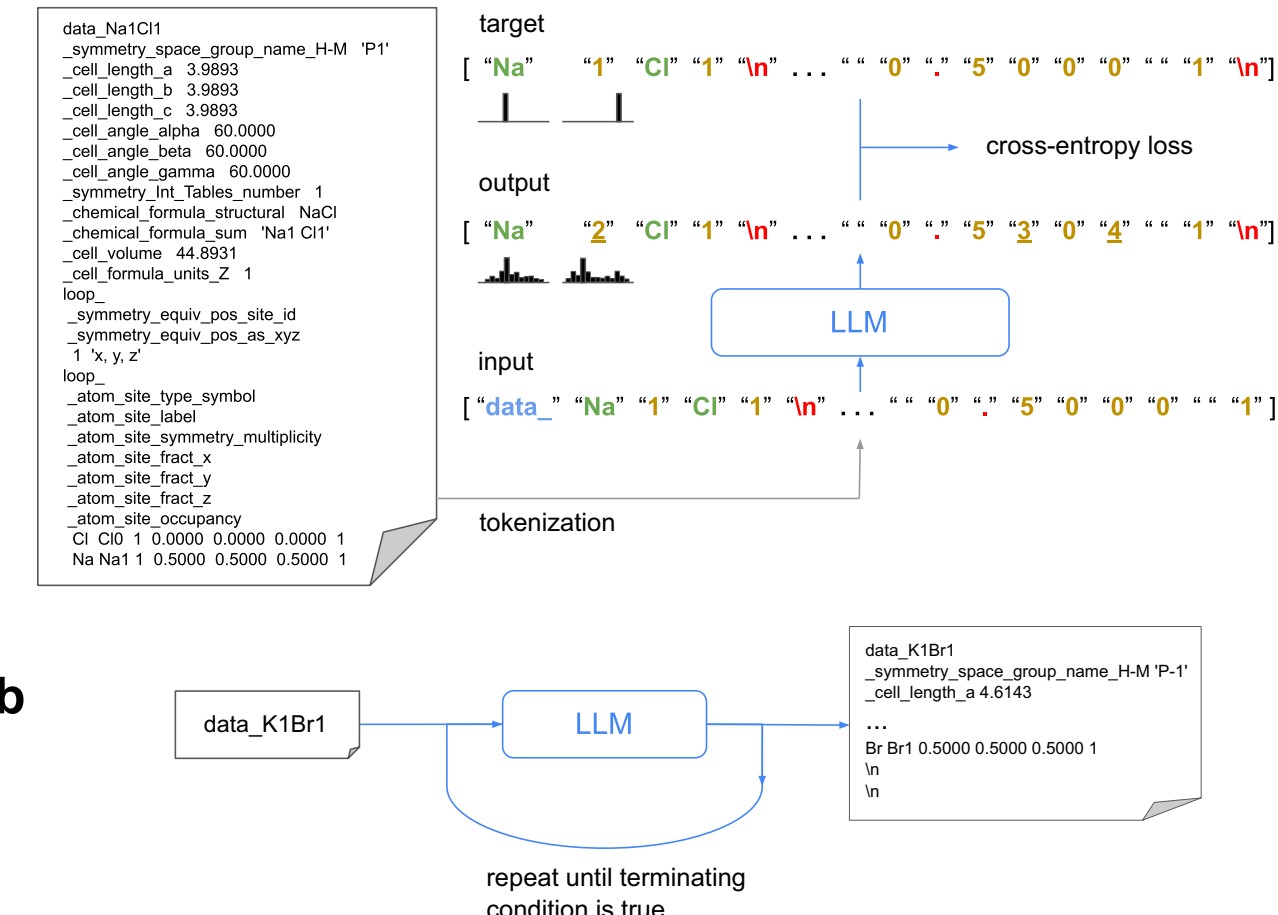

**Fig. 1 | Large language modeling of CIF files. a** Core concepts in training the model: A CIF file (left) is converted into a sequence of symbols, through tokenization. The sequence is processed by the model, which produces a list of probability distributions over the vocabulary, for each corresponding symbol in the input. The resulting predicted probability distributions are evaluated against the target distributions (which contain the entire probability mass on the correct subsequent token), using the cross-entropy loss metric. The target tokens are the input tokens shifted one spot to the left, as the objective is to predict the next token given a sequence of preceding tokens. The tokens are categorized as CIF tags (blue), atoms (green), numeric digits (gold), and punctuation (red). Output tokens (not actually sampled during training) represent the tokens assigned the highest probability by the model. Underlined tokens represent predicted distributions assigning a relatively low probability to the correct next token. **b** Generation of a CIF file: First, a prompt is constructed by concatenating the symbol `data_` with the desired cell composition, which is then tokenized and processed by the model. Next, a token is sampled from the predicted distribution for the upcoming token in the sequence. Finally, the sampled token is added to the accumulating contents of the CIF file. This procedure continues iteratively until a predefined terminating condition is met (e.g., two consecutive newline tokens are sampled).

the cell composition of the structure. Subsequently, the model is prompted with both the structure's cell composition and space group and again allowed to generate up to 3000 tokens. The prompt includes the first several lines of the pre-processed CIF file, up to the line containing the specification of the space group. Prompting the model with both the cell composition and space group allows us to assess how reliant the model is on the space group. This process is repeated for all CIF files of the held-out test set (10,286 in total).

The generated CIF files are then assessed for correctness and quality. Any syntactically incorrect CIF files are declared invalid. Syntactically correct CIF files are subjected to further analysis, and are considered to be valid only if specific criteria are met, such as being consistent in terms of generated structure and declared space group, and having reasonable bond lengths (see Supplementary Note 3 for further details on the validation of generated CIF files). The results of evaluating the generation of the CIF files of the test set using the small model are presented in Table 1.

The CIF files generated by prompting the model with the cell composition and space group were compared to the corresponding CIF files of the test set using a structure matching algorithm. We found that in 88.1% of cases there was at least one match with a test set structure within three generation attempts. The fraction of matching on first attempt and the fraction of matching on all three attempts, are presented in Supplementary Table 2, where we also provide the corresponding matching fractions for formulas that were not seen in training with any $Z$.

We further examined how closely the generated cell parameters resembled the actual cell parameters, for the cases where there was a structural match. We took the first matching structure for samples that had at least one generated structure matching the test set structure, and measured the $R^2$ and mean absolute error (MAE) for the true versus generated cell lengths, the true versus generated (i.e., printed) volume, and the implied (from generated cell parameters) versus generated volume. The results are presented in Fig. 2.

**Table 1 | Performance of the small model on the held-out test set**

|  | No Space Group | With Space Group |
|---|---|---|
| Space Group Consistent | 98.8% | 99.1% |
| Atom Site Multiplicity Consistent | 99.4% | 99.4% |
| Bond Length Reasonableness Score | 0.988 ± 0.069 | 0.988 ± 0.067 |
| Bond Lengths Reasonable | 94.6% | 94.6% |
| Valid | 93.8% | 94.0% |
| Longest Valid Generated Length | 1145 | 970 |
| Average Valid Generated Length | 331.9 ± 42.6 | 339.0 ± 41.4 |

The percentages represent the fraction of test set compounds which meet the corresponding criteria. For example, the first row represents the percentage of test set compounds where the declared space group in the generated CIF file is consistent with the generated structure. Valid generated length refers to the length of a valid generated CIF file in terms of the number of tokens.

To further assess the model's ability to generalize to unseen structures, we prompted the model with the cell compositions of the challenge set. The challenge set contains 58 structures not seen in training. These structures were all manually sourced from the recent literature, and represent experimentally characterized materials. Crucially, these compounds originate through a process different from the process which generated the training set (namely, a high-throughput DFT analysis of hypothetical materials). They also represent a variety of different structural classes, such as intermetallics, silicates, sulfides and selenides, borates, phosphates, carbonates, and complex mixed-anion compounds.

Both the small and large models were prompted with the cell compositions of the challenge set, both with and without the space group. A total of 100 attempts were made to generate a structure from the given cell composition (and optionally space group). We record the successful generation rate, representing the fraction of compounds where at least one valid CIF file was generated in the 100 attempts, and the true match rate, representing the fraction of compounds where there was a structural match between a valid generated structure and the true structure reported in the literature. The results are presented in Table 2 and Supplementary Tables 3–6.

The results in Table 2 indicate that inclusion of the space group in the prompt increases the likelihood of generating a valid structure, and of generating a match with the true structure. The large model appears to be superior to the small model in all categories. While the models can recover the reported structure more often when the structure was seen in training, it is noteworthy that they are able to generate unseen structures which match the reported structure in up to 40% of the cases.

**Comparison with other ML-based approaches**

Generative models of materials based on advanced ML techniques have been developed recently, some concurrently with this work. Due to the unavailability of source code and complete benchmarking results for all these emerging models, conducting an in-depth comparison between the approaches remains challenging. Nevertheless, here we present a comparison with other ML-based approaches. CDVAE[14], DiffCSP[40], DiffCSP++[41], and UniMat[42] are examples of diffusion-based approaches, whereas Gruver et al.[43] introduced a fine-tuned version of the LLaMA-2 model[44] for crystal structure generation. DiffCSP focuses on CSP through an equivariant diffusion process, while CDVAE uses a diffusion-based approach within a variational autoencoder framework for generating periodic materials. DiffCSP++ augments the equivariant diffusion process by introducing support for space-group constrained generation, through the incorporation of

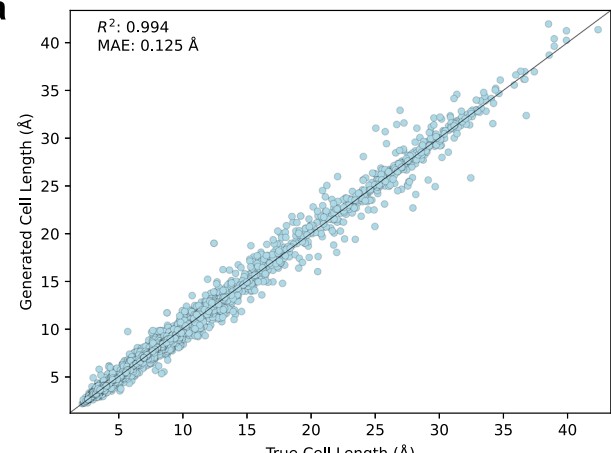

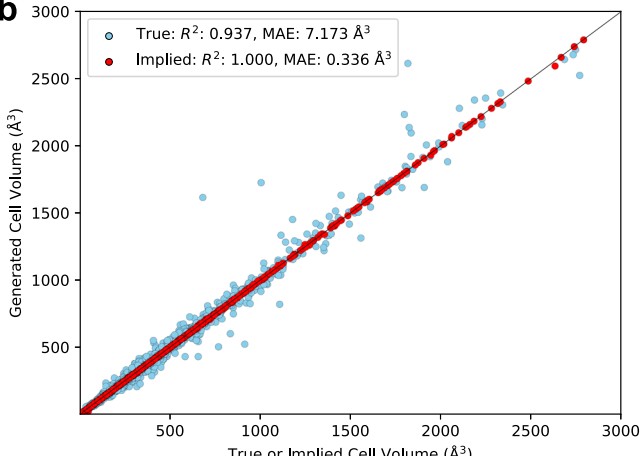

**Fig. 2 | Generated vs. true or implied cell parameters. a** The generated cell lengths for matching structures of the test set vs. the true cell lengths, when space group is included. **b** The generated cell volumes for matching structures of the test set vs. either the true cell volumes, or the cell volumes implied from the generated cell parameters, when space group is included. Source data are provided as a Source Data file.

prior knowledge of Wyckoff positions which constrain the diffusion process. UniMat re-purposes the 3D U-Net architecture[45,46] for unconditional generation, and generation conditioned on composition, and is trained on a large dataset of millions of structures.

We compare CrystaLLM to these models in both conditional and unconditional generation settings. In the (CSP, or conditional) generation setting, we compare CrystaLLM to these models on four benchmarks: Perov-5[47,48], Carbon-24[49], MP-20[50], and MPTS-52[51]. The Perov-5 dataset consists of 18,928 perovskites, Carbon-24 consists of 10,153 carbon allotropes, MP-20 consists of 45,231 stable inorganic materials of various classes, while MPTS-52 consists of 40,476 various inorganic materials. MPTS-52 is by far the most complex dataset, with up to 52 atoms in the unit cells of the constituent structures. In the unconditional generation setting, CrystaLLM is compared to these models on the Perov-5, Carbon-24, and MP-20 benchmarks.

The benchmark datasets have each been split into training, validation and test sets. All models are trained solely on the training set. For the CSP task, the models are used to generate 20 structures for each of the cell compositions of the test set. The models are evaluated in terms of the match rate, which is the fraction of compositions for which the true structure was generated within $n$ attempts (we tried $n = 1$ and 20), and the average root mean squared error (RMSE) of the

**Table 2 | Results of the small and large models on the challenge set, both with a space group (s.g.) and without**

|  | Small model | | Large model | |
| --- | --- | --- | --- | --- |
|  | no s.g. | with s.g. | no s.g. | with s.g. |
| Successful Generation Rate | 85.7% | 88.6% | 87.1% | 91.4% |
| Match Rate (Seen) | 50.0% | 50.0% | 83.3% | 83.3% |
| Match Rate (Unseen) | 25.9% | 34.5% | 37.9% | 41.4% |

The first row represents the percentage of cases where the model was able to generate a valid structure within 100 attempts. The second row represents the percentage of cases where a generated structure matched the true structure, for the compounds seen in training. The last row represents the percentage of cases where a generated structure matched the true structure, for unseen compounds only.

closest candidate for each test set structure. For the unconditional generation task, the models are given 10,000 generation attempts, and are evaluated in terms of metrics such as validity rate and coverage. In the validity tests, following the metrics presented in other studies, a structure is defined as valid if no interatomic distances are below 0.5 Å, and a composition is defined as valid if a charge neutral combination of the constituent atoms in the generated stoichiometry is possible. Coverage measures how closely the generated materials match the distribution of ground truth materials. Coverage precision is a measure of how many generated materials are a close match to materials from the ground truth set and is an indication of the quality of the generated materials. Being a close match is defined by distance between the materials with a pre-defined metric (see Supplementary Note 5 for more details). Coverage recall measures how many of the ground truth materials are matched by at least one generated material, and is a measure of how diverse the generated materials are. For example, a generating process could have high COV-P by simply generating the same valid material each time, but COV-R would be low in this instance. The AM(S/C)D measures are similar to coverage statistics, but measure the minimum distance between a generated material and the materials in the ground truth set; these measures are also separated across structural matching (AMSD) and composition matching (AMCD). While the COV-R and COV-P metrics have become established for the purposes of evaluating generative models of materials, we note that they must be interpreted cautiously, as they have several drawbacks. Primarily, the metrics do not fully account for the novelty of the generated materials, focusing instead on similarity, which depends on arbitrarily set thresholds. This can favor models which are overfit to the dataset, and not necessarily generalizable. Moreover, the metrics can be sensitive to the relative sizes of the test and generated sets, which can lead to potentially misleading scores, since a larger generated set together with a smaller test set might result in artificially high COV-R values, while a smaller generated set could inflate COV-P values.

The results for the CSP task are presented in Table 3, including the performance of three different versions of CrystaLLM. Versions $a$ and $b$ are trained on the benchmark data only and differ in the size of the model used. Version $c$ is trained on the full 2.3M training points minus the test set of MPTS-52 and is included to demonstrate how the results improve with the size of training data, but is not directly comparable to other models due to the different training data sets.

In the CSP task, CrystaLLM outperforms DiffCSP on three out of four benchmarks in terms of RMSE for both $n = 20$ and $n = 1$, and in terms of match rate when constrained to only a single generation attempt. This is achieved even in the most challenging of the benchmarks, MPTS-52, which contains structures with larger unit cells and more atoms.

For the unconditional generation task, the results for both the small and large CrystaLLM models, with different sampling temperatures are given in Supplementary Table 7. CrystaLLM is competitive with the other models on this task, and also achieves strong results in

terms of compositional validity on MP-20 and Perov-5, and obtains the highest COV-P value on Carbon-24. Furthermore, the best AMSD metrics are achieved by CrystaLLM on all three benchmarks.

CrystaLLM has important advantages when compared to the other models. In comparison to the diffusion-based methods, CrystaLLM supports both conditional and unconditional generation seamlessly, without requiring any architectural adjustments. More (or less) information is simply provided in the prompt, accordingly. Conversely, DiffCSP requires architectural augmentation to support unconditional generation, and CDVAE also requires an architectural adjustment to support conditional generation. Another important advantage is that CrystaLLM natively supports space-group constrained generation, with no changes or external processing required. Conversely, DiffCSP++ was devised as a separate approach dedicated to handling space-group constrained generation. It relies on a template retrieval and substitution method when the space group is unknown. In contrast, CrystaLLM generates a suitable space group automatically, with no extra work required. The DiffCSP++ template-based approach consequently makes it difficult to propose structures when no suitable template exists, which is a limitation that CrystaLLM does not have. CDVAE and UniMat do not support space group-constrained generation.

In comparison to the fine-tuned LLaMA-2 model, the largest CrystaLLM model has 200 million parameters, whereas the smallest fine-tuned LLaMA-2 model has 7 billion parameters, a difference of more than an order of magnitude in the number of parameters. The smaller size of CrystaLLM makes it easier to deploy for inference tasks, and much more accessible for training and fine-tuning. Additionally, while the fine-tuned LLaMA-2 model supports the constructs of natural language in its prompts, the flexibility of its inputs suggests that CrystaLLM may be conditioned on other properties of the structure as well, including those not traditionally included in the CIF format.

Finally, as a neural language model, CrystaLLM can leverage the established practice of fine-tuning, allowing the pre-trained model to be adapted for the prediction of materials properties. There is far less precedent in fine-tuning models based on diffusion and variational autoencoder architectures for tasks involving regression or classification.

The differences above between CrystaLLM and previous methods indicate that CrystaLLM has the unique advantage of being a more flexible, general-purpose model, capable of supporting a number of different generation use cases, without requiring a switch between architectural variants, or different models entirely, and which can be deployed in a cost-effective manner. CrystaLLM can alternate seamlessly between unconditional generation (when neither composition nor space group is known), generation conditioned on composition only, and generation conditioned on both composition and space group. Notably, it supports the conditioning of structure generation on specific symmetry space groups without being restricted, in principle, to the availability of known templates, a capability unique to CrystaLLM.

## Examples of generated structures

To further examine the model's ability to generalize to unseen scenarios, we prompted the model with various formulas, and examined its output. The results are presented in Fig. 3.

An example of the model generalizing to a formula that had been seen in training, but with different space groups, is presented in Fig. 3a. The formula, $Ba_2MnCr$, was in the held-out test set, with the R-3m space group. That combination of formula and space group had not been seen in training. The model generated a structure matching the one in the test set on the first attempt, when the space group was provided.

The model also demonstrated the ability to generate plausible structures for formulas not seen in training with any $Z$. An example is

**Table 3 | Benchmark CSP results, with *n* representing the number of samples generated for each structure of the benchmark test set.*** The CDVAE and DiffCSP results are taken from ref. [40]

| Model | *n* | Perov-5 | | Carbon-24 | | MP-20 | | MPTS-52 | |
|---|---|---|---|---|---|---|---|---|---|
| | | Match Rate | RMSE | Match Rate | RMSE | Match Rate | RMSE | Match Rate | RMSE |
| CDVAE | 1 | 45.31 | 0.1138 | 17.09 | 0.2969 | 33.90 | 0.1045 | 5.34 | 0.2106 |
| CDVAE | 20 | 88.51 | 0.0464 | 88.37 | 0.2286 | 66.95 | 0.1026 | 20.79 | 0.2085 |
| DiffCSP | 1 | *52.02* | *0.0760* | 17.54 | 0.2759 | 51.49 | 0.0631 | 12.19 | 0.1786 |
| DiffCSP | 20 | **98.60** | **0.0128** | **88.47** | 0.2192 | **77.93** | 0.0492 | **34.02** | 0.1749 |
| CrystaLLM [a] | 1 | 47.95 | 0.0966 | *21.13* | *0.1687* | 55.85 | 0.0437 | 17.47 | 0.1113 |
| CrystaLLM [a] | 20 | 98.26 | 0.0236 | 83.60 | 0.1523 | 75.14 | 0.0395 | 32.98 | 0.1197 |
| CrystaLLM [b] | 1 | 46.10 | 0.0953 | 20.25 | 0.1761 | *58.70* | *0.0408* | *19.21* | *0.1110* |
| CrystaLLM [b] | 20 | 97.60 | 0.0249 | 85.17 | **0.1514** | 73.97 | **0.0349** | 33.75 | **0.1059** |
| CrystaLLM [c] | 1 | - | - | - | - | - | - | 28.30 | 0.0850 |
| CrystaLLM [c] | 20 | - | - | - | - | - | - | 47.45 | 0.0780 |

*Numbers in bold indicate the best *n* = 20 result, while numbers in italics represents the best *n* = 1 result, amongst the models trained only on the benchmark training sets.
[a]Results for the small model architecture trained only on the benchmark training set.
[b]Results for the large model architecture trained only on the benchmark training sets.
[c]Results for the small model architecture trained on the original 2.3M-structure dataset without the structures of the MPTS-52 validation or test sets.

the quaternary compound CsCuTePt. This compound was not in the training set, but was in the held-out test set (with $Z = 4$). The model generated a structure matching the one in the test set, in the $F\bar{4}3m$ space group, on the third attempt when the space group was provided. The generated structure is presented in Fig. 3b.

Figure 3c shows the generated structure of $YbMn_6Sn_6$[52], an example of the model generalizing to structural motifs with elements not seen in training. This formula was not seen in training for any $Z$, and was not in the held-out test set. However, $ZrMn_6Sn_6$ was seen in training, in the P6/mmm space group. The model generated a structure in the same space group on the first attempt, without the space group being provided. The generated structure matched the $ZrMn_6Sn_6$ structure, with Yb substituted for Zr, and with cell parameters and atomic coordinates adjusted accordingly. This demonstrates the model performing a structure prediction by analogy procedure, as commonly used by materials scientists for discovery[53,54], despite never having been provided with the procedure to do this.

We now discuss the performance of the model in the context of four widely known classes crystal structures: rutiles, spinels, elpasolites, and pyrochlores.

Rutiles are a class of binary compounds that adopt a tetragonal unit cell, in the P4$_2$/mnm space group ($Z = 2$), as is seen in $TiO_2$, from which this class of materials adopts its name. The general formula for rutile oxides is $MO_2$, where M is a metallic species in the +4 oxidation state. Rutile fluorides are also known, where the metal is in the +2 oxidation state. The model's training dataset consisted of essentially all of the rutiles one might expect to be able to find in nature. Therefore, to test the model's ability to generate unseen rutiles, we requested the generation of theoretically possible, but unlikely compounds, such as $AuO_2$. With gold in a highly unlikely +4 oxidation state, $AuO_2$ is not expected to be formed under most conditions. However, the model was able to imagine what the structure of such a compound might be (when the space group is provided). While $TiO_2$ has cell parameters $a = 4.594$ Å, $c = 2.959$ Å, the generated rutile gold variant has $a = 4.838$ Å $c = 3.429$ Å, reflecting the increased volume occupied by the larger gold atoms (Fig. 3d).

Spinels are a group of ternary compounds with general formula $AB_2X_4$. The most common combination of elements in spinels is one where A is a cation in the +2 oxidation state, B is a cation in the +3 oxidation state, and X, normally a chalcogen, is a −2 anion. Spinels form cubic close-packed structures, with eight tetrahedral, and four octahedral sites, normally in the $Fd\bar{3}m$ space group. To explore the model's ability to generate unseen spinels, we selected the thiospinel $Sm_2BS_4$, which was absent from both the training and test sets. The

model was able to generate the expected spinel structure when the cell composition and space group were provided (Fig. 3e). During training, the model encountered a number of different oxy-, thio-, and selenospinels, and this likely contributed to its ability to generate this compound.

Elpasolites are quaternary compounds with the general formula $ABC_2X_6$. The A and C species are typically alkali metal cations in the +1 oxidation state, B is usually a transition metal cation in the +3 oxidation state, and X is a halogen anion. The elpasolites are often referred to as "double perovskites", since their structures are related to perovskites by the doubling of their unit cell dimensions, and the replacement of the divalent cation with alternating monovalent and trivalent cations. Elpasolites crystallize in the $Fm\bar{3}m$ space group, and are the most common quaternary crystal system reported in the Inorganic Crystal Structure Database[55]. We wondered if the CrystaLLM model could generate elpasolites not seen during training. We selected an elpasolite from the held-out test, that was not seen in training: the fluoride $KRb_2TiF_6$. The model was able to generate the correct elpasolite structure when the cell composition and space group was provided (Fig. 3f).

Finally, we considered pyrochlores, for which the general formula is $A_2B_2O_7$. Here A, a trivalent cation, and B, a tetravalent cation, are either rare-earths or transition metals (other oxidation states, e.g., combining monovalent and pentavalent cations, are also possible, but we focus here on the trivalent/tetravalent pyrochlores). Pyrochlores crystallize in the $Fd\bar{3}m$ space group ($Z = 8$). There are many combinations of A and B that are possible for this structure, by using lanthanide ions, actinide ions, and Y(III) for the A species, and various transition metal ions, as well as Ti(IV), Zr(IV), and Hf(IV) for the B species. We investigated whether CrystaLLM could generate valid pyrochlore structures for any unseen combinations, and whether it could estimate reasonable cell parameters in line with the trends observed for the pyrochlore series, as the cell parameters are expected to be correlated with the ionic radii of the A and B cations.

We created a space of pyrochlores consisting of 144 compounds by producing different combinations of A and B species. Of these, 54 were seen in training. We selected 10 compounds from among the 90 not seen in training, and attempted 3 generations with the model, for each. The cell composition and space group were included in the prompt. All generations resulted in valid pyrochlore structures. We subsequently performed DFT relaxation calculations on the first generated structure for each of the 10 compounds. One case, $Ce_2V_2O_7$, posed challenges in calculation under the generalized gradient approximation and was thus excluded from further analysis. The DFT-

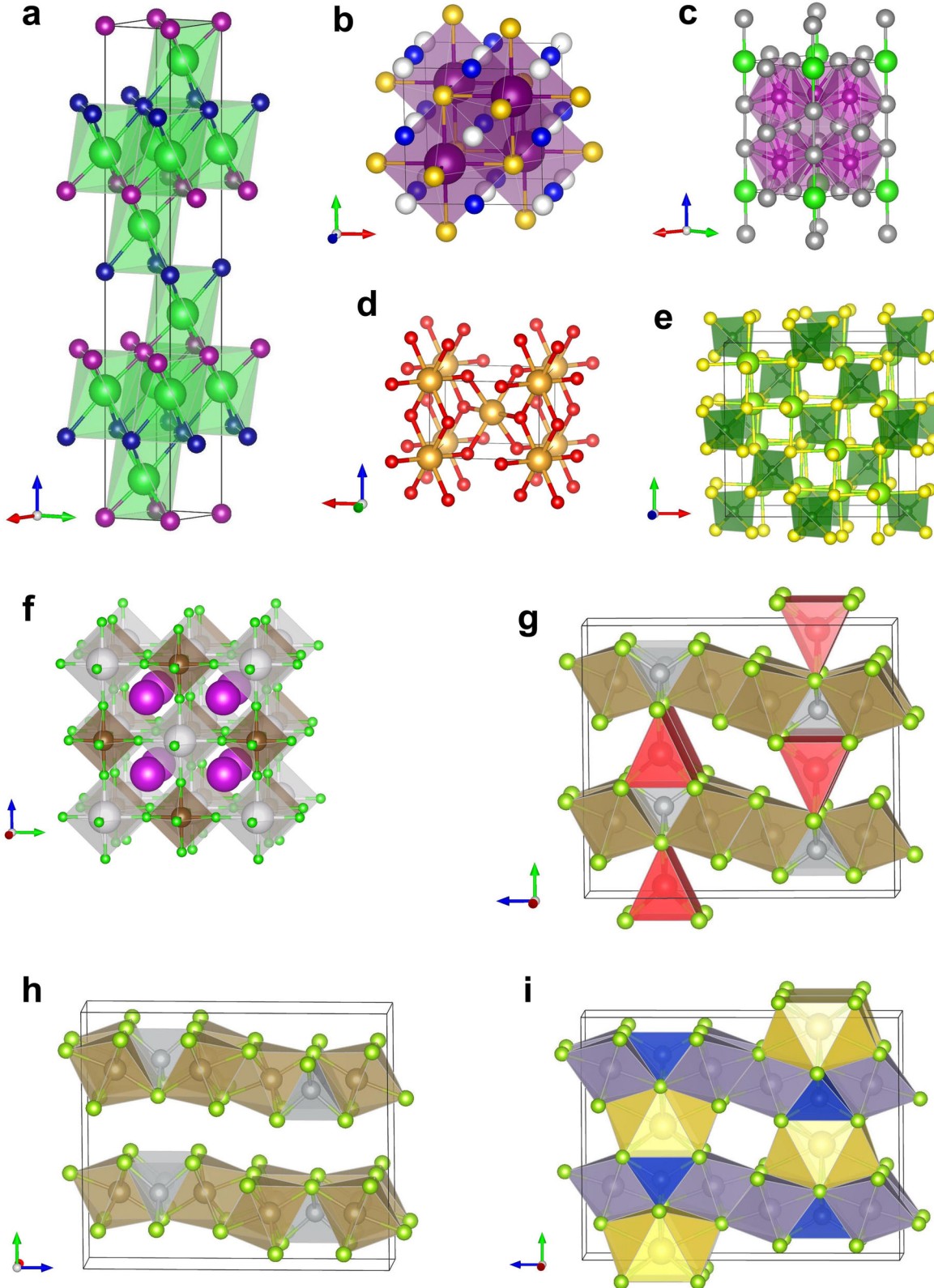

**Fig. 3 | The generated structures of various inorganic compounds. a** Ba$_2$MnCr. Cell parameters: *a, b*: 3.778 Å, *c*: 27.503 Å, *α, β*: 90.0°, *γ*: 120.0°. Color scheme: Ba: green, Mn: purple, Cr: blue. **b** CsCuTePt. Cell parameters: *a, b, c*: 7.153 Å, *α, β, γ*: 90.0°. Color scheme: Cs: purple, Cu: blue, Te: gold, Pt: white. **c** YbMn$_6$Sn$_6$. Cell parameters: *a, b*: 5.488 Å, *c*: 8.832 Å, *α, β*: 90.0°, *γ*: 120.0°. ZrMn$_6$Sn$_6$, in the training set, possessed the same structure, but with the following cell parameters: *a, b*: 5.364 Å, *c*: 8.933 Å, *α, β*: 90.0°, *γ*: 120.0°. Color scheme: Yb: green, Mn: magenta, Sn: gray. **d** AuO$_2$. Cell parameters: *a, b*: 4.838 Å, *c*: 3.429 Å, *α, β, γ*: 90.0°. Color scheme: Au: yellow, O: red. **e** Sm$_2$BS$_4$. Cell parameters: *a, b, c*: 10.884 Å, *α, β, γ*: 90.0°. Color scheme: Sm: light green, B: green, S: yellow. **f** KRb$_2$TiF$_6$. Cell parameters: *a, b, c*: 8.688 Å, *α, β, γ*: 90.0°. Color scheme: K: white, Rb: purple, Ti: brown, F: green. **g** LiTa$_2$NiSe$_5$ (*a*: 3.517 Å, *b*: 13.362 Å, *c*: 15.156 Å), which resembles the recently reported structure in[63]. **h** Ta$_2$NiSe$_5$, seen in training. **i** NaSn$_2$CuSe$_5$, seen in training. Source data are provided as CIF files in the Source Data file.

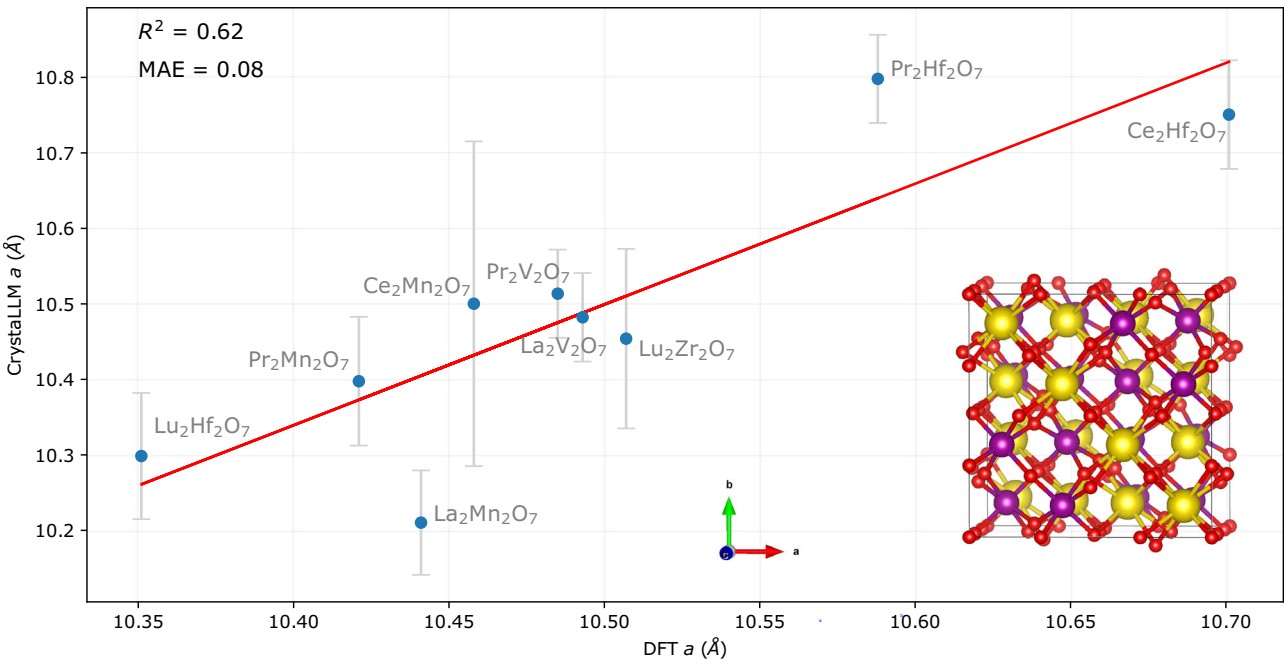

**Fig. 4 | Pyrochlore case study results.** Generated vs. DFT-derived values of the cell parameter $a$ for selected pyrochlores not in the training dataset. The error bars represent the $\pm$ standard deviation of the value of the $a$ cell parameter for the three generation attempts (all of which resulted in the pyrochlore structure), while the $y$-coordinate of the points represents the mean value of the cell parameter across the three attempts. The inset represents the structure of the generated pyrochlore $Pr_2Mn_2O_7$, with cell parameters $a$, $b$, $c$: 10.34 Å, $\alpha$, $\beta$, $\gamma$: 90.0°. Color scheme: Pr = yellow, Mn = purple, O = red. Source data are provided as a Source Data file.

derived value of the cell parameter for each of the remaining compounds is plotted against the mean value generated by CrystaLLM in Fig. 4. A good agreement exists between the DFT-derived and generated cell lengths, with an $R^2$ of 0.62 and MAE of 0.08 Å being exhibited. This example illustrates CrystaLLM's capability to accurately estimate cell parameters of compounds not seen in training with any structure.

While the model seems capable of generating structures for many different classes of inorganic crystals, it does nonetheless have difficulty in certain cases. All of the cases appear to involve systems that are rare, and under-represented in the training dataset, or missing from the training set altogether. More precisely, we define a template as a unique combination of the reduced composition ratio, the space group, and $Z$. For example, the combination of the reduced composition ratio 1:1:3:4, space group Cmcm, and $Z = 4$, represents a unique template. There are 25,921 unique templates in the dataset.

The problematic cases in the challenge set are largely represented by unseen templates, and templates for which there are few examples. For example, validation rates were low for $Mg_7Pt_4Ge_4$, the structure of which was reported recently to exist in the $P6_3mc$ space group ($Z = 2$)[56]. In this case, there were only 38 examples of 7:4:4 systems in the training dataset, none contained Mg or Pt, and none were in the $P6_3mc$ space group.

The small version of the model also seems to struggle with generating phosphates, sulfates, carbonates, and organic-inorganic hybrid structures. Examples include carbonate hydroxide minerals, such as $Co_2CO_3(OH)_2$[57] and $Cu_2CO_3(OH)_2$ (malachite). While present in the dataset, they belong to a group of analogous structures for which there are only a handful of examples. While both the small and large versions of the model can generate $Mn_4(PO_4)_3$, they generally fail to generate a valid structure for $Ca_5(PO_4)_3(OH)$ (hydroxyapatite). A common theme is the appearance of multiple oxyanions, which can give rise to more complex arrangements of atoms, for which the model may not have seen enough examples. In contrast, the model can generate compounds of the perovskite class reliably. However, over 5000 examples of the $ABX_3$ (X = O, F) system in the $Pm\bar{3}m$ space group were seen in

training. Finally, structures represented by CIF files with a relatively large number of tokens also pose challenges for the models. Future versions of the model will consider strategies for addressing these occurrences of class imbalance.

### Heuristic search for low-energy structures

The examples generated in the previous section were produced through top-$k$ random sampling of the model. Essentially, as the CIF file is generated, each subsequent token is sampled randomly from amongst the top $k$ tokens, according to their probabilities. (See Supplementary Note 2.4 for a detailed description of top-$k$ sampling.) However, random sampling may not necessarily result in the most desirable sequence, and consequently, there are more strategic approaches for constructing sequences that incorporate the probability distributions produced by the model, along with additional heuristics. An example of a heuristic search is Beam Search[58], which is commonly used in natural language contexts to improve the quality of generated sequences. Another popular heuristic search algorithm is MCTS, which has traditionally been used in the context of planning and games, but has recently also been used to increase the quality of generated natural language, through incorporation with LLMs[59].

Here, we employ the MCTS algorithm, informed by CrystaLLM, to generate a collection of sequences, which is expected to progressively yield sequences of increasingly higher quality as the search advances. In this implementation, each node in the tree represents a cumulative context of tokens. The algorithm operates through a series of steps, including selection, expansion, rollout, evaluation, and backpropagation. The search tree is constructed iteratively, as the search proceeds (Fig. 5). In the selection phase, nodes are chosen using the PUCT algorithm (Predictor-Upper Confidence bound applied to Trees)[60,61], which is a principled means of obtaining a balance between exploring untried nodes, and exploiting promising nodes. The expansion involves adding child nodes based on predicted probabilities. During the rollout step, the CrystaLLM model is prompted with token sequences until a terminating condition is met, leading to

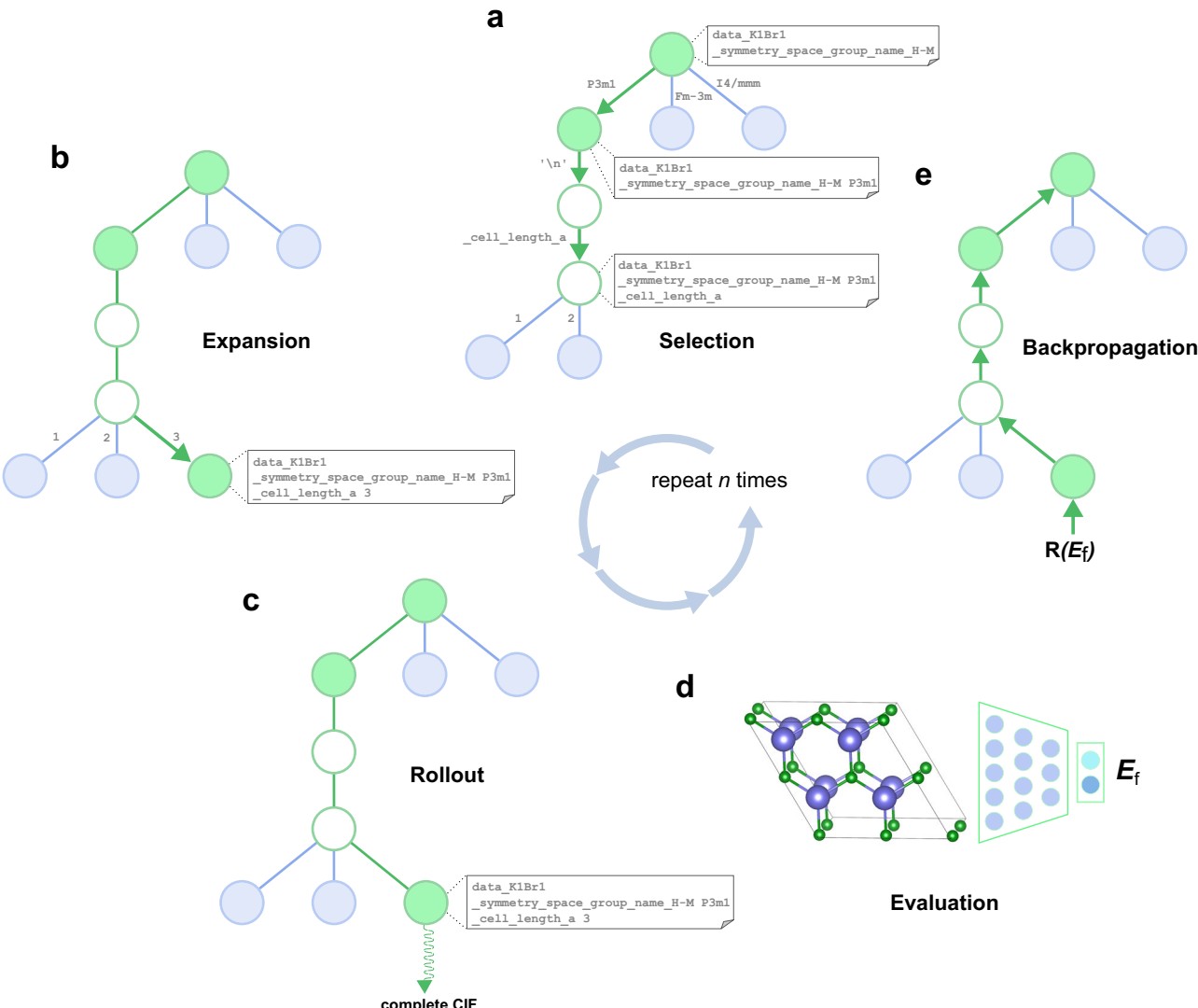

**Fig. 5 | The Monte Carlo Tree Search decoding procedure.** CIF files are generated as a tree is iteratively constructed, with each iteration guiding the generation of subsequent structures towards more desirable parameters (e.g., lower formation energy per atom). The nodes in the tree represent the cumulative contents of a CIF file at various points. **a** The Selection step involves descending the tree by choosing the most promising node at each level, using a variant of the PUCT algorithm. **b** During Expansion, an unexplored child node is randomly selected and added to the tree. If a node has only one highly probable child (represented as empty nodes), the child node bypasses the Rollout step. **c** The Rollout step involves prompting the model with the contents of the selected node, and sampling from the model until a terminal condition is met, so as to obtain a complete CIF file and an estimate of the value of a node. **d** The generated structure is validated and scored, incorporating the prediction of the structure's formation energy per atom, as given by a pre-trained neural network. **e** Finally, the score is backpropagated through the selected nodes, which store the accumulated results of each iteration. The resulting generated CIF file, if valid, is returned.

the evaluation of the completed sequence. Evaluation is conducted using the ALIGNN (Atomistic Line Graph Neural Network) model of formation energy per atom[62], while the backpropagation step accumulates outcomes in the tree nodes, scoring each based on the quality of the generated structure. (See Supplementary Note 4 for a more detailed description of the algorithm.) The objective is to produce structures with lower formation energy per atom, $E_f$, and the incorporation of the ALIGNN model allows for a fast and sufficiently accurate estimate of the target property.

When compared to random sampling, MCTS improves the overall validity rate for a compound, and also generally produces lower energy structures. To evaluate the MCTS decoding procedure, we took the 20 most problematic cases of the challenge set where the validity rate was greater than 0, and performed 1000 generation attempts using random top-$k$ sampling, and 1000 iterations of MCTS. The results are presented in Table 4.

When no space group is provided in the prompt, the validity rate improves in 95% of the cases, and the minimum $E_f$ attained improves in 85% of cases. (See Supplementary Tables 8 and 9 for more detailed results.) In some cases, the validity rate increases as the search proceeds when using MCTS (see Supplementary Fig. 6).

To further test the performance of MCTS, we applied the procedure to 102 novel compounds generated unconditionally by CrystaLLM (see the following section "Generating Novel Materials" for details of the unconditional generation). On these materials, we performed MCTS decoding with 1000 iterations each, using ALIGNN to provide feedback. After MCTS, the ALIGNN energy decreased (or remained constant) for all the compositions, with an average energy change of −153 ± 15 meV/atom (compared to the structures generated without MCTS). The mean $E_{hull}$ for the 102 structures, as calculated by DFT, improved by −56 ± 15 meV/atom on average, to 0.34 eV/atom; 22 of those structures were within 0.1 eV/atom of the hull. Further

demonstration of the statistical significance of ALIGNN-based MCTS, and details of the results, are provided in Supplementary Note 6. Future improvements of the energy estimators will increase the effectiveness of the MCTS approach.

## Generating novel materials

The discovery of novel and stable compounds can expand the capabilities of materials science. To understand the potential of using CrystaLLM for generating novel and feasible crystalline solids, we used the large model trained on the 2.3M-structure dataset to generate 1000 structures unconditionally, and assessed the stability of the novel compounds among them, using DFT. Of the 1000 generated CIF files, 900 were valid, and 891 represented structurally distinct (i.e., unique) materials. There were 102 structures which were novel when compared to the training dataset (established using structure matching). We performed DFT relaxation of the 102 novel structures, and compared the energy of each structure with the convex hull as given by the Materials Project. The mean $E_{hull}$ of the 102 novel structures was

0.40 eV/atom. Notably, we found that 20 structures were within 0.1 eV/atom of the hull, including 3 with $E_{hull}$ = 0.00 eV/atom. Fig. 6 depicts the 4 most stable of the novel compounds. (See Supplementary Table 10 for comprehensive results for the 20 most stable compounds.)

Inspection of the novel materials revealed that the model generated a mix of ionic, semi-ionic, and metallic compounds. The compounds with lower energy above the hull tended to be ionic and semi-ionic in nature. This could be due to the model being better at learning the coordination rules of ionic and semi-ionic compounds, as they are typically more defined and stricter than those for metallic compounds. For example, in most oxides, Fe will be coordinated tetrahedrally or octahedrally to oxygen. For metallic compounds, it is less defined, a priori, what the coordination patterns should be. In fact, many metallic compounds only stabilize due to disorder thanks to the configurational entropy (effects which are not considered here). The model has, therefore, a better chance of generating a stable ionic material than a stable ordered metallic compound.

## Beyond element substitution

Although CrystaLLM appears to be very effective at finding appropriate template systems for a given cell composition, and making the necessary adjustments of cell parameters to substitute different atoms, it appears capable of going further, synthesizing information from different template systems. An example is the selenide LiTa₂NiSe₅, which is obtained by lithium intercalation into Ta₂NiSe₅[63].

The compound LiTa₂NiSe₅ was not present in the training set; however, the layered material Ta₂NiSe₅ was (Fig. 3g, h). As LiTa₂NiSe₅ was included in the challenge set, we performed 100 generation attempts with the model. While the model was not able to recover the lowest energy structure reported, it did produce structures with close

**Table 4 | Results of MCTS decoding for the 20 most problematic cases of the challenge set**

|  | No Space Group | With Space Group |
|---|---|---|
| Validity Rate Improvement | 95.0% | 60.0% |
| Minimum $E_f$ Improvement | 85.0% | 65.0% |
| Mean $E_f$ Improvement | 70.0% | 65.0% |

The percentages represent the fraction of cases with the corresponding improvement after using MCTS decoding, when compared to random sampling. The first row represents the percentage of cases where the validity rate improved. The second row represents the percentage of cases where the minimum $E_f$ obtained was improved. The third row represents the percentage of cases where the mean $E_f$ was improved.

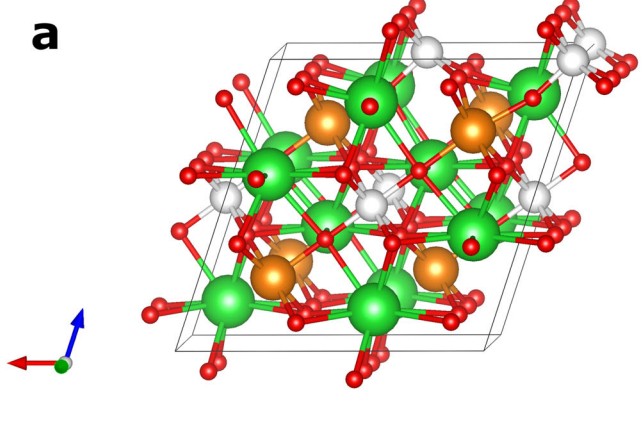

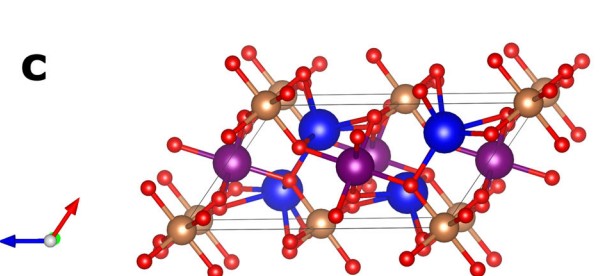

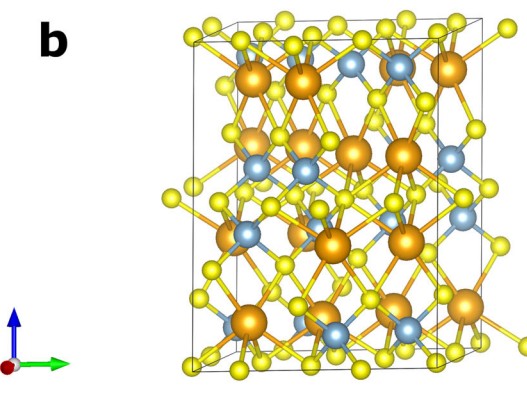

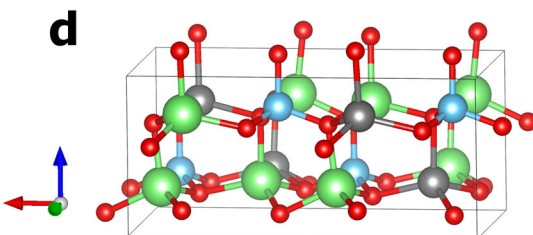

**Fig. 6 | Unconditionally generated novel structures.** The four lowest-energy novel structures generated unconditionally by the large model. **a** Ba₄Na₂Ir₂O₁₁ $Z$ = 2, Cm. Cell parameters: $a$: 10.308 Å, $b$: 5.995 Å, $c$: 10.269 Å, $\alpha$, $\gamma$: 90.0°, $\beta$: 108.5°. Color scheme: Ba: green, Na: orange, Ir: white, O: red. $E_{hull}$ = 0.00 eV/atom. **b** NaAlS₂ $Z$ = 16, P2₁. Cell parameters: $a$: 10.233, $b$: 10.277 Å, $c$: 13.703 Å, $\alpha$, $\gamma$: 90.0°, $\beta$: 100.9°. Color scheme: Na: orange, Al: gray, S: yellow. $E_{hull}$ = 0.00 eV/atom. **c** Ca₂YSbO₆ $Z$ = 2, P2₁/c.

Cell parameters: $a$: 5.651 Å, $b$: 5.853 Å, $c$: 9.850 Å, $\alpha$, $\gamma$: 90.0°, $\beta$: 125.0°. Color scheme: Ca: blue, Y: purple, Sb: bronze, O: red. $E_{hull}$ = 0.00 eV/atom. **d** Li₂FeSiO₄ $Z$ = 4, Pna2₁. Cell parameters: $a$: 10.988 Å, $b$: 6.278 Å, $c$: 5.026 Å, $\alpha$, $\beta$, $\gamma$: 90.0°. Color Scheme: Si: light blue, Fe: dark gray, Li: light green, O: red. $E_{hull}$ = 0.02 eV/atom. Source data are provided as CIF files in the Source Data file.

resemblance to low-energy polymorphs. Upon closer examination of the dataset, we found that $NaSn_2CuSe_5$ was present (Fig. 3i), which likely provided some precedent for the intercalation of atoms between layered structures. It thus appears that the model is capable of integrating information from different template systems to form new structural predictions.

## The CrystaLLM.com web application

To allow for easy and open access to the CrystaLLM model, we make it available through a web application, published at https://crystallm.com/. The application allows users to enter in a reduced formula, and optionally a value for $Z$ and the desired space group. The option to select the model size is also provided. The request is sent to the model, and the resulting structure (or the CIF contents, if the structure is invalid) is presented to the user. By making the model easily accessible, we hope to contribute a potentially useful tool to the materials structure research community. We also hope to receive feedback from users that may help improve future versions of the model.

## Discussion

Here, we have shown that LLMs of the CIF format are able to generate inorganic crystal structures for a variety of known classes. Indeed, the model is able to produce valid and sensible arrangements of atoms in 3-dimensional space by generating $xyz$ coordinates digit-by-digit. The model also seems to have captured the relationship between space group symbols and the symmetries inherent in the structures it generates.

We chose to build a language model of the CIF format (instead of a simplified format, for example, which might include a minimal vocabulary) for several reasons. First, the CIF format is not particularly verbose. The model learns the grammatical structure of the format fairly quickly. We can thus avoid having to devise an intermediate format that requires inter-conversion between more common formats, which could also be error prone. Second, we believe that having the model learn to generate the more redundant parts of the CIF format, such as the cell volume, and $Z$, which are inferable from prior inputs, helps the model to perform better overall.

A number of approaches for crystal structure generation have been reported[64–67]. These approaches generally require the existence of pre-defined structural templates, and are followed by the procedural or ML-assisted substitution of atoms and adjustment of cell parameters, under the constraint of a specified space group. These types of approaches can also be enhanced to increase the structural diversity of generated materials, by allowing partial substitutions and adjusting substitution probabilities[68]. Conversely, CrystaLLM automatically selects the templates which can be applied to a given composition, utilizing the implicit templates it has absorbed through autoregressive training. Moreover, the model can automatically adjust cell parameters to accommodate the atoms in the unit cell. It can also produce structures based on templates it has not explicitly encountered in training, borrowing from its internalized concepts of chemical structure. In comparison with recently reported diffusion-based ML methods for crystal generation (CDVAE[14] and DiffCSP[40]), not only does CrystaLLM outperform them on established benchmarks in several aspects, but it also offers additional advantages in terms of flexibility (e.g., in using symmetry as input) and the potential for fine-tuning.

While the CrystaLLM model can generate sensible structures, this does not by itself make it suitable, as is, for CSP. Just as natural language LLMs, such as GPT-3 and -4, are not suitable chatbots without further fine-tuning and alignment, the CrystaLLM model will also need to be fine-tuned for more advanced tasks. Fine-tuning involves an additional and separate training step, where the model's parameters are adjusted in the context of a different task. This may also involve altering the model's output layer, such as to make it suitable for a regression task. Models can be fine-tuned using a variety of techniques, but supervised learning and reinforcement learning[69] are most common. One might use reinforcement learning, for example, when a task is not clearly defined as a supervised learning problem. When fine-tuning natural language LLMs for chatbot applications, it is common to use Reinforcement Learning from Human Feedback (RLHF)[70,71]. With RLHF, the idea is to gather data from human annotators to be used to train a reward model, which scores generated text according to its desirability. The reward model is then used as part of a reinforcement learning-based tuning of the LLM. In CSP, one would like to produce ground-state structures (for some given physical conditions). One could thus imagine an analogous procedure where CrystaLLM is fine-tuned for the goal of generating low-energy structures, via feedback from an external evaluator of the generated structure's energy, resulting in what we may call Reinforcement Learning from Thermodynamic Feedback. This procedure would also require a reward model, and such a model should ideally provide a timely estimate of a structure's energy. This excludes time-consuming approaches such as DFT. A viable approach could make use of a separate ML-based model of formation energy, such as one based on ALIGNN. Indeed, neural network potentials have been used to accelerate the prediction of crystal structures, and the identification of potentially stable materials[72,73].

There are several limitations with the current approach. First, none of the structures of the dataset have site-occupancy disorder (fractional site occupancies). Therefore, CrystaLLM cannot generate disordered structures, and may not successfully generate structures for combinations of cell composition and space group that imply a disordered structure. An example is $K_2NaTiOF_5$, which is reported to be an elpasolite, in the $Fm\bar{3}m$ space group ($Z = 4$), with F and O species sharing the same crystal site[74]. Another limitation is that the CIF files of the dataset were not all created using the same level of theory. The training set is derived from a combination of DFT sources using different settings, functionals, etc., which may make it difficult for the model, in some instances, to learn a consistent relationship between cell composition and detailed structure[75].

Nevertheless, we believe that CrystaLLM will be a useful tool for crystal structure generation, which is quickly becoming a critical step in largescale materials discovery[68,76], and materials informatics. We plan to explore fine-tuning the model for physical property prediction tasks, such as the prediction of lattice thermal conductivity, where experimental data is relatively scarce[77]. The architecture of the model allows it to be fine-tuned for either composition-based or structure-based prediction tasks. This implies that CrystaLLM may be the basis for a general-purpose materials informatics model, which can be used for generative tasks, and fine-tuned for property prediction tasks that require either composition or structure. If the model is able to transfer what it has learned about the world of atoms to these various predictive problems, it may prove to be a quite flexible tool relevant to many aspects of materials chemistry.

## Methods
### Dataset curation

The dataset was assembled by obtaining structures from the Materials Project[50], the OQMD[78], and NOMAD[79], which were originally optimized using density functional theory (DFT) simulations. Specifically, the structures from the Materials Project were downloaded in April 2022, and from NOMAD in April 2023. We use version 1.5 of the OQMD, which was released in October 2021. In total, ~3.6 million structures were obtained. This dataset consists of compounds containing anywhere from 1 to 10 elements, with most consisting of 3 or 4 elements. The elements up to and including atomic number 94 are present, with the exception of polonium, astatine, radon, francium, and radium. The dataset contains roughly 800,000 unique formulas, and 1.2 million unique cell compositions. When paired with space groups, there are 2.3 million unique cell composition-space group pairs. (See Supplementary Fig. 1.) To choose between duplicate structures containing the

same cell composition and space group, the structure with the lowest volume per formula unit was selected. The 2.3 million structures in this dataset were converted to CIF files using the pymatgen library[80], and were used for training. The CIF files were created with the pymatgen option for symmetry finding tolerance set to 0.1 Å. All floating point numbers in the files were rounded to 4 decimal places. The dataset was split randomly into train, validation, and test sets, such that the training set consisted of 2,047,889 CIF files, the validation set 227,544 CIF files, and the test set 10,286 CIF files.

### CIF syntax standardization and tokenization
The dataset of CIF files was standardized and tokenized prior to training. The vocabulary consisted of CIF tags, space group symbols, element symbols, numeric digits, and various punctuation symbols, for a total of 371 symbols. After tokenization, the training set consisted of 768 million tokens. See Supplementary Note 1 for further details.

### Generative pre-training
The generative pre-training step requires a vocabulary, $\mathcal{V}$, and an ordered list of tokens $\mathcal{U} = (u_1, \ldots, u_n)$, with $u_i \in \mathcal{V}$. We want to maximize the following likelihood:

$$\mathcal{L}(\theta; \mathcal{U}) = \sum_i \log P(u_i | u_{i-c}, \ldots, u_{i-1}; \theta) \tag{1}$$

where $c$ is the size of a context window, $P$ is the conditional probability distribution to be modeled, and $\theta$ the parameters of a neural network. We therefore minimize $\mathcal{J}(\theta; \mathcal{U}) = -\mathcal{L}$, using stochastic gradient descent to adjust the parameters. We use a multi-layer Transformer decoder[81] for the neural network, as described in ref. [20]. Our model consists of 25 million parameters, with 8 layers, 8 attention heads, and an embedding size of 512. We decay the learning rate from $10^{-3}$ to $10^{-4}$ over the course of training, and use a batch size of 32. For further details, see Supplementary Note 2.

### Evaluation of generated structures
A CIF file is said to be valid if: (1) the declared space group is consistent with the generated structure, (2) the generated bond lengths are reasonable, and (3) the declared atom site multiplicity is consistent with the cell composition. To check if the generated structure is consistent with the printed space group, we use the `SpacegroupAnalyzer` class of the pymatgen library, which uses the spglib library[82]. To check if bond lengths are reasonable, we first use a Voronoi-based nearest-neighbor algorithm in pymatgen to identify bonded atoms; then, we establish expected bond lengths based on the electronegativity difference between the bonded atoms, and their ionic or covalent radii. We classify a structure as having reasonable bond lengths if all the detected bond lengths are within 30% of the corresponding expected bond lengths. See Supplementary Note 3 for more details on how the validity of a generated CIF file is established.

In some scenarios, we wish to determine whether a generated structure matches a target structure, which typically represents a ground-truth structure. To determine whether two structures are a match, we use the pymatgen `StructureMatcher` class, which performs a structural similarity assessment of two crystals. We use a fractional length tolerance of 0.2, a site tolerance of 0.3 Å, and an angle tolerance of 5 degrees, which are the default values in pymatgen. Both structures are reduced to primitive cells before matching, and are scaled to equivalent volume.

### Benchmark evaluations
To evaluate CrystaLLM on the Perov-5, Carbon-24, MP-20, and MPTS-52 benchmarks, we consider two different scenarios: (1) the model is trained only on the benchmark training sets, and (2) the model is trained on the full 2.3 million-structure dataset minus the validation

and test set structures of the MPTS-52 dataset. For the first scenario, both the small and large model architectures are used. We use the same 60-20-20 train/validation/test splits used in the CDVAE study[14] for the Perov-5, Carbon-24, and MP-20 datasets, and we use the same 27,380/5,000/8,096 train/validation/test split used in the DiffCSP study for the MPTS-52 dataset. These models are trained for a fixed number of iterations: the Perov-5 model is trained for 1750 iterations, the Carbon-24 model is trained for 8000 iterations, the MP-20 model is trained for 5000 iterations, and the MPTS-52 model is trained for 3500 iterations. For the second scenario, we train a model with the small model architecture on the full 2.3 million-structure dataset minus the structures of the MPTS-52 validation and test sets. The model is trained for 100,000 iterations. We decay the learning rate from $10^{-3}$ to $10^{-4}$ over the course of training, and use a batch size of 32, for all models. For both scenarios, we take the structures of the test set(s), and prompt the models with only the cell compositions of these structures. Models are given 20 attempts to generate a structure. We use top-$k$ sampling with $k = 10$ and a temperature of 1.0 for all models and in both scenarios.

To establish the match rate and RMSE, we use the same procedure defined in the DiffCSP study. Specifically, we use the pymatgen `StructureMatcher` class, with a fractional length tolerance of 0.3, a site tolerance of 0.5 Å, and an angle tolerance of 10 degrees, to determine if a generation attempt matches the ground truth structure. The RMSE, normalized by $\sqrt[3]{V/N}$ (where $V$ is the volume of the lattice and $N$ is the number of sites), is computed between the corresponding ground truth structure and each matching generated structure. The test set's average RMSE is computed by taking the lowest RMSE for each entry's matching generated structure.

To evaluate CrystaLLM on the unconditional generation tasks, we train a model on the training sets of each of the Perov-5, Carbon-24 and MP-20 datasets, using both the small and large model architectures. We use the same 60-20-20 train/validation/test splits used in the CDVAE study[14]. These models are trained for a fixed number of iterations: the Perov-5 model is trained for 5000 iterations, the Carbon-24 model is trained for 8000 iterations, and the MP-20 model is trained for 5000 iterations. We decay the learning rate from $10^{-3}$ to $10^{-4}$ over the course of training, and use a batch size of 32, for all models. Models are then given 10,000 generation attempts, starting from the prompt 'data_'. Each generation attempt results in both a generated cell composition and structure. We use top-$k$ sampling with $k = 30$ and temperatures of 0.5 and 0.7 for all models.

To establish the unconditional generation metrics, we follow the same procedure defined in the CDVAE study. Specifically, structural fingerprints are created using the `CrystalNNFingerprint` class with the "ops" preset, and compositional fingerprints are created using the `ElementProperty` class with the "magpie" preset, both provided by the matminer library[83]. For the coverage metrics, we use the standard cutoff values: for MP-20, we use a structure cutoff of 0.4 and a composition cutoff of 10; for Carbon-24 and Perov-5, we use a structure cutoff of 0.2 and a composition cutoff of 4.

### Monte Carlo tree search decoding
The MCTS search tree is constructed iteratively, as the search proceeds. We maintain a tree width of 5 and maximum tree depth of 1000. The PUCT constant $c_{puct}$ is set at 1.0. The expansion involves adding child nodes based on predicted probabilities. When a node has a probability of 0.99 or greater, it becomes the only child node, and bypasses the rollout step. During the rollout step, the CrystaLLM model is prompted with token sequences until a terminating condition is met, up to a maximum of 1000 tokens. Evaluation is conducted using the ALIGNN model of formation energy per atom. The ALIGNN model is given the generated CIF file, and the predicted formation energy per atom (in eV) is used to compute the reward. The backpropagation step accumulates outcomes in the tree nodes, scoring each based on the

quality of the generated structure, with a reward constant $\lambda$ of 2.0. For all compounds, we perform 1000 search iterations. See Supplementary Note 4 for a more detailed description of the algorithm.

### Uniqueness and novelty of generated materials

To assess the model's ability to generate materials unseen in training, the model is prompted with '`data_`' 1000 times, each resulting in a CIF file. We use top-$k$ sampling with $k = 10$ and a temperature of 1.0. (In principle, the chosen temperature should affect the trade off between novelty rate and how reasonable the generated structures are, so temperature should be considered a parameter to be optimized in future studies.) To establish uniqueness and novelty of the generated structures, we use the pymatgen `StructureMatcher` class, with a fractional length tolerance of 0.2, a site tolerance of 0.3 Å, and an angle tolerance of 5 degrees. A generated compound is considered unique if it represents a structural type that appears only once amongst all compounds generated during the experiment, under the specified tolerances for lattice dimensions and atomic positions configured for the `StructureMatcher` class. A generated compound is considered novel if it is structurally distinct from all of the compounds in the dataset used to train the model.

### DFT calculations

For the pyrochlore case study, a small number of DFT calculations were performed using VASP, following as closely as possible the settings used in the OQMD project (where most of the pyrochlore structures seen in training were taken from). For example, the recommended PAW potential was used for each element: Zr_sv for zirconium, Hf_pv for hafnium, Lu_3 for lutetium, Pr_3 for praseodymium, Ce_3 for cerium (for the remaining elements, the name of the PAW potential simply matched the element's symbol). The Perdew-Burke-Ernzerhof (PBE) exchange-correlation functional[84], in the generalized-gradient approximation, was used in all calculations. Hubbard (PBE+U) corrections were applied for transition metal elements with unfilled d levels ($U_{eff}$ = 3.8 eV for Mn and 3.1 eV for V). Although the cell parameters reported here correspond to the conventional cubic cell with 8 formula units, the DFT calculations were performed using the primitive cell with two formula units, and sampling of the reciprocal space corresponding to that primitive cell was performed using a $7 \times 7 \times 7$ grid, as done for all pyrochlore calculations in the OQMD project.

For the DFT calculation of the energy against hull of the unconditionally generated compounds, we also used the VASP code, following the Materials Project settings[85], i.e., same functional (PBE), $U_{eff}$ parameters, PAW potentials, etc. to ensure compatibility with reference compounds in the hull. Structures generated with CrystaLLM were relaxed to the nearest local minima within the generated unit cell, without symmetry constraints on the atomic coordinates (we applied small random displacements of less than 0.1 Å to the initial coordinates). All the DFT calculations converged, electronically and ionically, within the standard convergence thresholds in the Materials Project setup.

### Web application

The web application is made available at https://crystallm.com. The user of the application is presented with a text field requiring a formula to be entered. Optionally, they may provide the number of formula units ($Z$), the desired space group, and the size of the model. Once they press the `Generate` button, a request is sent to a GPU server which has the model in memory. The request is converted into a prompt, and the generated contents are returned to the user. If no $Z$ is provided, we scan through $Z$ values of 1, 2, 3, 4, 6, and 8, and return the first valid structure generated by the model. We validate the generated structure using the same procedure described previously, checking that the generated structure is consistent in terms of the printed space group, and other elements of the CIF file. If no valid structure can be found, the user is presented with an informative error message, including the option to view the generated content. Requests typically take several seconds to process, but can take longer if no $Z$ is provided and the model has trouble generating a valid structure for the attempted $Z$ values. Generated structures are displayed in a web browser-based 3D structure viewer provided by the Crystal Toolkit framework, upon which the front-end of the web application is built[86].

### Reporting summary

Further information on research design is available in the Nature Portfolio Reporting Summary linked to this article.

## Data availability

Source data are provided with this paper. All trained models, training sets, and artifacts generated by the models have been deposited to Zenodo, and the files are publicly accessible[87] under the CC-BY 4.0 license. The structures used in the experiments described in this work were obtained from the Materials Project (https://materialsproject.org/), the OQMD, and NOMAD. All structures were made available by those sources under the Creative Commons Attribution 4.0 License[88]. Source data are provided with this paper.

## Code availability

The code for training and using the CrystaLLM model is open source, released under the MIT License. The code repository[89] is accessible online at: https://github.com/lantunes/CrystaLLM.

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

## Acknowledgements

This work was partially supported by computational resource donations from Amazon Web Services through the AWS Activate program, obtained with assistance from the Communitech Hub (L.M.A.). For the DFT calculations, we used the Young supercomputer facility via the UK Materials and Molecular Modeling Hub, which is partially funded by EPSRC through grants EP/T022213/1 and EP/W032260/1 (R.G.-C.).

## Author contributions

L.M.A. conceived the project, performed the experiments, and drafted the manuscript. L.M.A. and R.G.-C. designed the experiments. R.G.-C. carried out the DFT calculations. R.G.-C. and K.T.B. supervised and guided the project. All authors reviewed, edited and approved the manuscript.

## Competing interests

The authors declare no competing interests.
