## [Transparent Peer Review file · Nature Communications]

Crystal Structure Generation with Autoregressive Large Language Modeling

Corresponding Author: Mr Luis Antunes

Version 1:

Reviewer comments:

Reviewer #1

(Remarks to the Author)

This paper introduces a CrystaLLM model for crystal structure generation which is distinctively trained on textual representations standardized and tokenized from CIF format. The paper has constructive implications for the use of LLM in the generation of crystal structures in materials and provides a potentially usable Web application tool for the entire research community. However, several issues need to be refined before considering the manuscripts for acceptance.

1. Using a textual approach for data processing may result in the emergence of new equivalent sites due to changes in the number of elements, as exemplified by the output Na₂ in Figure 1.a. How does the model address the contradiction between the emergence of such new sites and the fixed length of the text?
2. Additional clarification is required regarding why Supported atom tokens and Supported space group tokens fail to cover all cases.
3. Further explanation is needed regarding the sources of the data. Out of the total 200+ million CIF files, how much does each database account for? Additionally, has any data filtering been applied to the databases used?
4. More training details need to be disclosed, such as the training time on small and large models. Furthermore, the article needs to disclose the evolution of the loss function during the model training process, to provide further insight into the behavior of the model.
5. Further elucidation is needed on the differences between using autoregressive LLM and other generative models for material generation. Here are some summaries of the applications of generative model in materials science that you can refer to:
[1] Generative Artificial Intelligence and Its Applications in Materials Science: Current Situation and Future Perspectives. <https://doi.org/10.1016/j.jmat.2023.05.001>.
[2] Generative models for inverse design of inorganic solid materials. <http://dx.doi.org/10.20517/jmi.2021.07>.
[3] Recent advances and applications of deep learning methods in materials science. <https://doi.org/10.1038/s41524-022-00734-6>.
6. The relationship between challenge set and training set, verification set and test set needs to be further clarified.
7. Additional clarification is needed regarding the meanings of All column and Reduced Unseen column in Table 2.
8. In Table 6, why does the Validity Rate increase more when the prompt does not include space group information?
9. It is recommended to provide a complete example of CIF file processing in the supplementary materials, including a comparison between the CIF files before and after processing, to enhance readers' understanding of the method.
10. The authors may benefit from further editing and optimization of both text and figures to enhance reader understanding and appreciation of the research.

Reviewer #2

(Remarks to the Author)

The paper "Crystal Structure Generation with Autoregressive Large Language Modeling" describes a language modeling approach toward generating CIF files for crystal structure prediction (CSP) and generation. The authors constructed a transformer language model trained from scratch on CIF files, with experiments on CSP benchmarks and case studies.

Overall, this reviewer finds the current paper premature for Nature Communications due to concerns over the conceptual contribution and the experimental validation presented.

At the time of this review, this reviewer is aware of two highly relevant papers that use language models:

[1] Flam-Shepherd, Daniel, and Alán Aspuru-Guzik. "Language models can generate molecules, materials, and protein binding sites directly in three dimensions as XYZ, CIF, and PDB files." arXiv preprint arXiv:2305.05708 (2023).

[2] Gruver, Nate, et al. "Fine-Tuned Language Models Generate Stable Inorganic Materials as Text." arXiv preprint arXiv:2402.04379 (2024).

These two papers and the current paper share the main method of using a language model for generating CIF files towards crystal generation. Compared to [1], the data curation, model design, benchmarking, and case studies are more comprehensive. [2] presents fine-tuned LLMs as opposed to training from scratch, with experimental/benchmarking results focused on discovering novel stable materials (as opposed to CSP benchmarks in the current paper). The current paper also presents an MCTS-based decoding scheme for generating low-energy structures.

In addition to the papers above, this reviewer also wants to point out two highly relevant papers that were publicly available at the time this review was written:

[3] Jiao, Rui, et al. "Space Group Constrained Crystal Generation." arXiv preprint arXiv:2402.03992 (2024).

[4] Zeni, Claudio, et al. "Mattergen: a generative model for inorganic materials design." arXiv preprint arXiv:2312.03687 (2023).

Both [3] and [4] are diffusion-based crystal generative models that are capable of space group conditional generation of crystal structures, and the overall capabilities are goals are highly relevant to the current paper. In light of the existing works and the relatively straightforward method of a transformer language model over CIF strings, this reviewer finds the current paper has limited conceptual contribution toward generative modeling of materials. While the MCTS decoding scheme is nice, experiments and analysis over this method are rather limited.

The CSP benchmarking results demonstrate the authors' method has comparable performance to DiffCSP and mostly better performance compared to CDVAE. The case studies are appreciated. However, the performance on the CSP benchmark does not notably advance the state-of-the-art. Further, this reviewer believes, that to establish the practical relevance of the current method, further experiments toward benchmarking the model's capability in generating novel and stable materials structures are needed. Such benchmarks are presented in [2], [4], and [5]:

[5] Yang, Mengjiao, et al. "Scalable diffusion for materials generation." arXiv preprint arXiv:2311.09235 (2023).

The benchmark procedure involves sampling from the generative model and evaluating the formation energy and the energy above the hull to validate if the generative model can discover novel, stable materials structures. Decent performance in the CSP task does not directly indicate the model's capability in discovering novel and stable materials structures, and to put the current model into context, a benchmark like those presented in [2,4,5] would be highly useful. This reviewer is not suggesting the current method has to outperform the methods presented in [2,4,5], but believes without such a benchmark it is hard to evaluate the practical value of the current model in materials.

This reviewer is a little confused by the authors' discussion on next-token prediction constituting a "world model". In particular, in the case of materials, the LM method breaks known physical symmetries such as the permutation of atoms. (~Line 45)

Reviewer #3

(Remarks to the Author)

Summary

This paper proposed a method to generate crystal structure using large language models with a refinement through Monte Carlo Tree Search. The authors meticulously trained a transformer decoder on millions of CIF files, employing autoregressive tasks. The proposed method's efficacy was evaluated across multiple benchmarks, and the authors have further provided an accessible web demo for demonstration.

Pros

The paper presents an exhaustive analysis of the generated samples categorized by space group. Both successful and challenging instances are thoroughly scrutinized, contributing to a comprehensive understanding of the model's capabilities.

Cons

The paper falls short in presenting convincing performance results. It lacks a demonstration of a substantial advantage over

previous diffusion-based methods regarding the quality of generated structures and computational efficiency. Crucial evaluations, such as the validity and coverage of randomly generated samples, are noticeably absent.

2. Lack of methodological comparison with related works. The idea of applying LLM to science domain generation has been explored in [1, 2]. While the concept of applying LLMs to science domain generation has been explored in prior research [1, 2], the paper fails to highlight fundamental distinctions between the proposed method and these existing approaches.

Additionally, compared to diffusion-based methods, the invariance of the modelled data distribution remains unexplored.

3. Some claims were not well supported. The abstract asserts that the proposed approach challenges conventional representations of crystals and demonstrates the potential of LLMs for learning effective 'world models' of crystal chemistry. However, there is an absence of discussion or results confirming the model's similarity to 'world models' or showcasing any significant 'world knowledge.' Given [2], the claim in the introduction, "Here, we report the first LLM specifically designed for crystal generation" was not accurate.

[1] Flam-Shepherd, D. & Aspuru-Guzik, A. Language models can generate molecules, materials, and protein binding sites directly in three dimensions as XYZ, CIF, and PDB files. arXiv preprint arXiv:2305.05708 (2023).

[2] Gruver, Nate, et al. "Fine-Tuned Language Models Generate Stable Inorganic Materials as Text." arXiv preprint arXiv:2402.04379 (2024).

Version 2:

Reviewer comments:

Reviewer #1

(Remarks to the Author)

Authors have revised the manuscript point by point. I recommend that the present version be accepted for publication.

Reviewer #2

(Remarks to the Author)

This reviewer thanks the authors for their detailed response and modification to the paper. The updated paper contains more discussion over related works and extends the experimental results to unconditional generation. These modifications definitely improve the paper.

Regarding conceptual novelty, this reviewer can agree with the authors that the findings in this paper are non-trivial and novel, based on Nature Communications' guidelines. This reviewer, however, still finds the experimental validation bears room for improvement.

While the non-conditional generation results are appreciated, as pointed out in the previous comments, this reviewer still feels some DFT-based validation is desired for a publication at Nature Communications. For the MCTS method, the benchmark was conducted over the 20 most problematic cases that were manually selected. Can the authors report performance improvement from the MCTS decoding method in a more general setting?

Reviewer #3

(Remarks to the Author)

The additional experimental results do not demonstrate a clear advantage of the proposed method over existing LLM-based and diffusion-based approaches. The contribution of this paper remains unclear, both in terms of experimental performance and as a technical solution to current challenges. The previous concerns have not been adequately addressed.

Version 3:

Reviewer comments:

Reviewer #2

(Remarks to the Author)

I appreciate the authors' efforts in adding DFT validation and additional experiments on the MCTS search algorithm. From the DFT validation results, the proposed model seems inferior to state-of-the-art models such as MatterGen. That said, this field is developing at a fast pace, and I appreciate the DFT results that put the model's performance in context.

I believe the evaluation for the MCTS search algorithm can be further improved as 102 is still a relatively small sample size. Since ALIGNN is used for getting formation energy signals, I encourage the authors to conduct a larger scale experiment and evaluate more aspects of the MCTS algorithm -- what is the error bar of the improvement brought by the search algorithm? Does the search algorithm reduce the diversity/novelty of the generated samples?

I in principle recommend paper acceptance.

Reviewer 1

This paper introduces a CrystaLLM model for crystal structure generation which is distinctively trained on textual representations standardized and tokenized from CIF format. The paper has constructive implications for the use of LLM in the generation of crystal structures in materials and provides a potentially usable Web application tool for the entire research community.

We thank the reviewer for the positive feedback, and agree that the implications for the use of LLMs in the generation of crystal structures are constructive. We also thank the reviewer for acknowledging the potential utility of the Web application for the entire research community.

However, several issues need to be refined before considering the manuscripts for acceptance.

1. Using a textual approach for data processing may result in the emergence of new equivalent sites due to changes in the number of elements, as exemplified by the output Na2 in Figure 1.a. How does the model address the contradiction between the emergence of such new sites and the fixed length of the text?

Figure 1a depicts the training procedure, where the model is tasked with predicting the token that follows a given token. In the example in the figure (i.e. during training), the model would simply be penalised for outputting a “2” instead of a “1”. On the other hand, during the generation (i.e. inference) procedure, the model is not limited to producing a fixed length of the text. The model will continue to generate new text until it generates a combination of tokens that signifies a break condition (see Figure 1b). As there is no limit to the length of text it can generate, if it generates a “2”, then it is free to generate the text representing more sites, as needed.

2. Additional clarification is required regarding why Supported atom tokens and Supported space group tokens fail to cover all cases.

The atom tokens cover all 89 atom types present in the training data. Atoms with atomic number $Z \geq 84$ (Po) are thus excluded (except for the early actinides Ac, Th, Pa, U, Np, and Pu, which did appear in crystal structures in the databases).

The 227 space group symbols also cover all space groups present in the training data. The space groups $P4_222$ (no. 93), $P6$ (no. 168), and $P432$ (no. 207) are not supported as there are no structures in the training data with these space groups. These three space groups are known to occur very rarely, due to a combination of symmetries (and absences of symmetries) that are difficult to realise in a crystal geometry [R1]. For example, the space group $P6$ requires the presence of a six-fold rotation axis but without the presence of mirror planes and inversion centres that occur in other hexagonal groups. Note that the rarity of these space groups is not limited to the *ab initio* databases used in our study. The ICSD database, which lists crystallographic information for most currently known materials and minerals, does not contain any experimentally-determined ordered inorganic compounds in these three space groups (only three experimental inorganic crystal structures are listed for these space groups: $K_2Ta_4O_9F_4$ and $MoCu_2Al_{7.92}$ for space group $P6$, and $Rb(NO_3)_3$ for space group $P432$, but they all exhibit site-occupancy disorder).

We have included this additional clarification in Supplementary Note 1 of the SI as requested by the reviewer.

3. Further explanation is needed regarding the sources of the data. Out of the total 200+ million CIF files, how much does each database account for? Additionally, has any data filtering been applied to the databases used?

We have provided a comprehensive breakdown of the sources of the training data in the supporting information. Some degree of filtering has been applied to deduplicate the datasets. The processing of the data has been detailed in the methods section ‘Data Curation’, and the distribution is depicted in Supplementary Figure 1. In our open source code repository, all of the code required to obtain and clean the data, as we did for the paper, is available.

4. More training details need to be disclosed, such as the training time on small and large models. Furthermore, the article needs to disclose the evolution of the loss function during the model training process, to provide further insight into the behavior of the model.

The loss as a function of training epoch is provided in the SI, in Supplementary Figure 2. We have added Supplementary Note 2.3 to the SI, which states the training times required for the small and large models.

5. Further elucidation is needed on the differences between using autoregressive LLM and other generative models for material generation. Here are some summaries of the applications of generative model in materials science that you can refer to:

[1] Generative Artificial Intelligence and Its Applications in Materials Science: Current Situation and Future Perspectives. <https://doi.org/10.1016/j.jmat.2023.05.001>.

[2] Generative models for inverse design of inorganic solid materials. <http://dx.doi.org/10.20517/jmi.2021.07>.

[3] Recent advances and applications of deep learning methods in materials science. <https://doi.org/10.1038/s41524-022-00734-6>.

We have added extensive benchmarking of the performance of CrystaLLM compared to other models, both LLMs and diffusion models. In discussing the results of this benchmarking, we include more detailed discussion of the differences between CrystaLLM and other models, as well as the relative strengths and weaknesses of different approaches. These additions can be found in section 2.3 of the main manuscript. We also thank the reviewer for bringing to our attention the important and relevant works in references [1-3]. We now cite these works in the Introduction section of the main manuscript.

6. The relationship between challenge set and training set, verification set and test set needs to be further clarified.

In our study, the train, test and validation sets have the standard definitions. That is, training sets are used to fit the parameters of the model, validation sets are used to compare trained models for model selection, and test sets are used to test the final resulting model. The train, test and validation sets are formed by random splitting of the same dataset, as detailed in the Methods section ‘Data Curation’, as such train, test and validation sets are independent and identically distributed. The challenge set is assembled from recent literature, to contain structures that are outside the training set, and also to include structures that are outside of the main distribution of the original set. The challenge set is described in more detail in section 2, page 3, of the main manuscript.

7. Additional clarification is needed regarding the meanings of All column and Reduced Unseen column in Table 2.

In Table 2, the difference between the “All” and “Reduced Unseen” columns is that the reduced compositions in “All” may be in the training set, but with a different Z . For example, Na_2Cl_2 may be in “All”, and Na_4Cl_4 may be in training. On the other hand, Na_2Cl_2 would not be in “Reduced Unseen”, since NaCl would occur in training with some other Z (i.e. $Z=4$). A footnote has been added to the caption of Table 2 to clarify this.

8. In Table 6, why does the Validity Rate increase more when the prompt does not include space group information?

Sometimes, including a certain combination of space group and composition may make generation of a valid structure more difficult. For example, if we provide just the formula NaCl , then the model is free to infer space group $Fm\bar{3}m$, but if we provide NaCl and $P6_3mc$ as space group, it might be more difficult to generate a valid structure based on the data that it has previously seen.

9. It is recommended to provide a complete example of CIF file processing in the supplementary materials, including a comparison between the CIF files before and after processing, to enhance readers’ understanding of the method.

In Supplementary Note 1 of the Supplementary Information we have added the original CIF file from which the example pre-processed CIF file was produced, to better illustrate the pre-processing step.

10. The authors may benefit from further editing and optimization of both text and figures to enhance reader understanding and appreciation of the research.

We have tried to be as clear as possible in our description of the work and have now added more context and comparison to other approaches. All figures in the manuscript are vector graphics, providing infinite resolution. We hope that this has clarified the study for the reviewer.

[R1] Urusov, V. S., & Nadezhina, T. N. (2009). Frequency distribution and selection of space groups in inorganic crystal chemistry. *Journal of Structural Chemistry*, 50, 22-37.

Reviewer 2

The paper “Crystal Structure Generation with Autoregressive Large Language Modeling” describes a language modeling approach toward generating CIF files for crystal structure prediction (CSP) and generation. The authors constructed a transformer language model trained from scratch on CIF files, with experiments on CSP benchmarks and case studies.

Overall, this reviewer finds the current paper premature for Nature Communications due to concerns over the conceptual contribution and the experimental validation presented.

We regret that the reviewer finds the work premature. However, we will argue below that the work represents, in fact, a unique and significant conceptual contribution to the nascent field of generative AI for materials. We have also gathered more experimental validation of the model’s capabilities, as detailed below. We believe this will show that the work is not premature for publication in Nature Communications.

At the time of this review, this reviewer is aware of two highly relevant papers that use language models:

[1] Flam-Shepherd, Daniel, and Alán Aspuru-Guzik. “Language models can generate molecules, materials, and protein binding sites directly in three dimensions as XYZ, CIF, and PDB files.” arXiv preprint arXiv:2305.05708 (2023).

[2] Gruver, Nate, et al. “Fine-Tuned Language Models Generate Stable Inorganic Materials as Text.” arXiv preprint arXiv:2402.04379 (2024).

We agree with the reviewer that these are two highly relevant papers, although still in pre-print stage, which according to Nature Communications’ guidelines, do not compromise the novelty of our work. Still, we have cited [1], stating in our paper that

a recent pre-print by Flam-Shepherd and Aspuru-Guzik, where the idea of generating the structures of molecules, materials, and protein binding sites with LLMs has been preliminarily explored

Moreover, we would like to make clear that our work pre-dates [2] and, in fact, our pre-print is cited by that work.

We discuss more about these two papers in the response to the point below.

These two papers and the current paper share the main method of using a language model for generating CIF files towards crystal generation. Compared to [1], the data curation, model design, benchmarking, and case studies are more comprehensive. [2] presents fine-tuned LLMs as opposed to training from scratch, with experimental/benchmarking results focused on discovering novel stable materials (as

opposed to CSP benchmarks in the current paper). The current paper also presents an MCTS-based decoding scheme for generating low-energy structures.

We agree with the reviewer’s comparison of our work with the cited papers. The MCTS-based decoding scheme we introduce is indeed a distinct contribution, which demonstrates the potential of alternate decoding approaches that [1] and [2] neglect. We would also like to highlight several more distinctions: Compared to [2], the largest CrystaLLM model has at least an order of magnitude fewer parameters than the smallest model in [2]. This makes training CrystaLLM from scratch practical and accessible, allowing for adaptations that are not possible through fine-tuning alone. The smaller size of CrystaLLM also makes the model much easier to deploy and use in generation tasks. While [2] describes using an A100 GPU throughout for inference, CrystaLLM can readily be used with an RTX 3090 GPU, which requires a fraction of the cost to deploy.

In terms of [1], note that that work appears preliminary in nature, and (at the time of this writing) many details regarding the training and experimental protocols are absent, including source code. In contrast, all code, datasets, and artifacts generated during experimentation for CrystaLLM are publicly available. Our model has been deployed via source code release, an API, and a web interface, and is being increasingly adopted by researchers in academia and industry. For example, the CrystaLLM web application and API have received over 100,000 requests combined since being released in August 2023, originating from over a dozen API keys issued to individual researchers, and from hundreds of web application users from various continents. The source code repository has been cloned actively since its public release in February 2024. This illustrates that our work does not follow from [1]; instead, both works emerged in parallel, and, in any case, our model is in a much more advanced stage of development and deployment.

In addition to the papers above, this reviewer also wants to point out two highly relevant papers that were publicly available at the time this review was written:

[3] Jiao, Rui, et al. “Space Group Constrained Crystal Generation.” arXiv preprint arXiv:2402.03992 (2024).

[4] Zeni, Claudio, et al. “Mattergen: a generative model for inorganic materials design.” arXiv preprint arXiv:2312.03687 (2023).

We acknowledge the importance of integrating all the relevant literature into the discussion of our manuscript. To explain why these references were missing in our submitted manuscript, we would like to clarify the timeline of these publications: Jiao et al. (2024) was uploaded to arXiv on February 6, 2024. As our manuscript was submitted to Nature Communications on December 22, 2023, this paper was not available at the time of our submission. Hence, it was not possible for us to consider this work during the preparation of our manuscript. Similarly, Zeni et al. (2023) was first available on arXiv on December 6, 2023. Given the proximity of this publication to our submission date, and the necessary lead time required for manuscript preparation and internal review, it was not feasible to include this work in our initial submission. Furthermore, it appears that neither of the cited works references our CrystaLLM pre-print, which was publicly available as a pre-print before both of these studies were posted (actually, since July 2023). This observation might suggest a mutual unawareness in the field due to the closeness in timing of these independent contributions.

Nevertheless, we now address these papers in our manuscript, including new benchmark comparisons, a discussion of different model architectures, and the benefits and drawbacks of the various approaches, in section 2.3 of the main manuscript. We have included direct comparisons of the performance of our model to these and other new models on benchmarking tasks where comparisons are available. Note the the model for [1] is not available for public use at this time, and the metrics presented in the pre-print are limited, thus extensive comparison as we have performed with other models is not possible.

Both [3] and [4] are diffusion-based crystal generative models that are capable of space group conditional generation of crystal structures, and the overall capabilities are goals are highly relevant to the current paper. In light of the existing works and the relatively straightforward method of a transformer language model over CIF strings, this reviewer finds the current paper has limited conceptual contribution toward generative modeling of materials.

We strongly disagree with the reviewer’s opinion that the manuscript has limited conceptual contribution. We have established that a (small) LLM, trained from scratch (i.e. not fine-tuned) on CIF files, can learn to generate unseen crystal structures with physical validity. We think this is a core contribution to the field, which will open new avenues for materials research, as evidenced by the papers already citing our pre-print. The fact that the method is straightforward (in the sense that it doesn’t involve complex physics - or no physics at all) only makes it more interesting. We consider it a strength of CrystaLLM that a relatively straightforward approach consistently performs at or above state-of-the-art for many of the benchmarks performed.

To illustrate the above point, we would like to highlight a recent review article (written by a team which includes Nobel prize winner Kostya Novoselov [R2]), where they state:

With the recent explosion of interest in language models (LM), two preprints [here they cite our CrystaLLM pre-print] suggest generating material structures as token sequences using Transformer [ref]. ... The authors claim performance comparable to physically-motivated state-of-the-art models, for generating both crystals and organic molecules. The LM don’t respect any kind invariance (permutation, translation, rotation), which makes these ongoing developments even more fascinating. Was the problem that simple all along?

As these authors do, we find the simplicity of our solution fascinating.

While the MCTS decoding scheme is nice, experiments and analysis over this method are rather limited.

We acknowledge that the reviewer appreciates the MCTS decoding scheme. However, in terms of the experiments and analysis over the MCTS method, which the reviewer considers “limited”, we are not clear about what the suggestion is. We already provide quite extensive analysis of the MCTS approach, including structure matching results on the challenge set, and also a plot of how the matching improves over the course of the search, in comparison to random sampling.

The CSP benchmarking results demonstrate the authors’ method has comparable performance to DiffCSP and mostly better performance compared to CDVAE. The case studies are appreciated. However, the performance on the CSP benchmark does not notably advance the state-of-the-art.

We did advance the state-of-the-art in the CSP task, as we generally surpassed DiffCSP in the $n=1$ results, and generally achieved the lowest RMSE numbers. Whether a result “notably” advances the state-of-the-art is a matter of opinion. Relatively smaller improvements are also often seen when models are performing close to the limit of what is possible on a given benchmark. For example, most models attain 100% structural validity (or very close to it) on the unconditional generation benchmark. This may be more a statement of the nature and usefulness of the benchmark rather than of the model itself. Furthermore, the value and novelty of our model go beyond the reported benchmark performance. For example, as argued above, the smaller size of CrystaLLM in comparison with some of the other models that have emerged recently, makes it much easier to train, deploy and use, and therefore much more likely to have an impact on the community.

Further, this reviewer believes, that to establish the practical relevance of the current method, further experiments toward benchmarking the model’s capability in generating novel and stable materials structures are needed. Such benchmarks are presented in [2], [4], and [5]:

[5] Yang, Mengjiao, et al. “Scalable diffusion for materials generation.” arXiv preprint (2023).

The benchmark procedure involves sampling from the generative model and evaluating the formation energy and the energy above the hull to validate if the generative model can discover novel, stable materials structures. Decent performance in the CSP task does not directly indicate the model’s capability in discovering novel and stable materials structures, and to put the current model into context, a benchmark like those presented in [2,4,5] would be highly useful. This reviewer is not suggesting the current method has to outperform the methods presented in [2,4,5], but believes without such a benchmark it is hard to evaluate the practical value of the current model in materials.

While we acknowledge the importance of evaluating formation energy and energy above the hull, we

believe that the current scope of our work, which focuses on the generative capabilities and structural validity of CrystaLLM, is an essential and valuable contribution on its own. The ability to generate structurally valid and chemically plausible materials is a crucial step in materials discovery, and serves as a strong foundation for future work that can include more extensive energy calculations using DFT.

Furthermore, the generative models in [2,4,5] have adopted various benchmarking methodologies, reflecting the diversity in approaches and the evolving nature of this field. Our current benchmarks, which include consistency checks and the CSP and unconditional generation tasks, align with established practices for initial validation of generative models.

Although CrystaLLM was not originally conceived as an unconditional generative model, we have nonetheless evaluated the model on a series of established benchmarks for unconditional generation, and now report results for the Perov-5, Carbon-24 and MP-20 benchmarks using metrics such as COV-R and COV-P. We found that CrystaLLM performs surprisingly well on these tasks, which highlights the flexibility of the approach. We therefore thank the reviewer for raising this issue and prompting us to test the model beyond its original intention. The performance on these benchmarks, comparison to other LLM models and diffusion models, and a discussion of the strengths and weaknesses of the different approaches is now in section 2.3 of the manuscript.

In future work, we plan to extend our evaluation to include formation energy and energy above the hull using DFT calculations, thereby providing a more comprehensive assessment of the stability and novelty of generated structures. However, we believe that our present results already demonstrate significant advancements, and provide a solid basis for further investigation. We reiterate that CrystaLLM demonstrates excellent performance on both the CSP and unconditional generation tasks, which represent foundational aspects of materials discovery.

This reviewer is a little confused by the authors' discussion on next-token prediction constituting a "world model". In particular, in the case of materials, the LM method breaks known physical symmetries such as the permutation of atoms. (~Line 45)

The model is indeed sensitive to the positions of atoms in the cell composition it is prompted with. This is because we standardize the order of the atoms in the cell composition by ordering them based on electronegativity. However, this does not in general imply that the LM method breaks known physical symmetries such as the permutation of atoms, or that a language model cannot learn a world model of crystal chemistry. In theory, the language model could be trained on different permutations of atoms in the cell composition to support such permutability in the prompt. However, this would significantly increase the size of the training set and the training requirements. Therefore, we chose to simplify the task by ensuring that the cell composition order is always standardized when the model is prompted. By standardizing the order, we ensure consistency and practicality in training and application, while still respecting the underlying physical symmetries. It is unrealistic to expect the model to discover the rule that atoms in a composition are permutable if it has not seen a single example demonstrating this. Though we believe that it can, in principle and in practice, learn such a rule, much as the popular natural language LLMs learn invariances in word position when shown ample examples.

There are strong suggestions that the model "understands" what it is generating, and that it is not simply capturing spurious correlations in the sequence of tokens. For example:

1. The mathematical correspondence between crystallographic cell parameters and cell volumes, evidenced by the perfect correlation between generated cell volumes and implied volume (from cell parameters), as shown in Figure 2b.
2. The consistency between Wyckoff positions, space group and composition, which is evidenced by the high validity rates of Table 1.
3. The constraints in terms of coordinates corresponding to reasonable bond lengths, again evidenced in Table 1.
4. The correlation between ionic sizes and the predicted cell parameters, for a given crystal structure; evidenced in the pyrochlore test case and tested against DFT.

We conjecture that if the model were simply learning surface statistics, it would have to have seen exponentially more examples in training to accomplish these same feats on unseen structures. Given the representational power of Transformer models with millions of parameters, it is entirely plausible to us that the task of next-token prediction could lead to the formation of a world model.

While we strongly believe that CrystaLLM does learn an internalized world model, we recognize that we have not performed experiments explicitly addressing this question. As such, we have removed the reference to ‘world models’ in our abstract. We still include a mention of world models in our introduction, but this is to motivate our work, and we do not claim that we have directly established that CrystaLLM has a world model of crystal chemistry. Proving or disproving the world model hypothesis would require demonstrating connections between the model’s internal representations and its outputs, and is far beyond the scope of our work. We nonetheless think that it is a very interesting conjecture about LLMs in general, so we would like to leave it in the introduction.

The codebase seems satisfactory.

We thank the reviewer for the feedback regarding the codebase. We have put considerable effort into ensuring that it is well-documented and accessible, and we appreciate acknowledgment of this aspect.

[R2] Rashid, A., Lazarev, M., Kazeev, N., Novoselov, K., & Ustyuzhanin, A. (2024). Review on automated 2D material design. *2D Materials*.

Reviewer 3

Summary

This paper proposed a method to generate crystal structure using large language models with a refinement through Monte Carlo Tree Search. The authors meticulously trained a transformer decoder on millions of CIF files, employing autoregressive tasks. The proposed method’s efficacy was evaluated across multiple benchmarks, and the authors have further provided an accessible web demo for demonstration.

We appreciate the reviewer’s summary, and positive recognition of our work. We have indeed put considerable effort into making the web application accessible, and we thank the reviewer for acknowledging this aspect of the work.

Pros

The paper presents an exhaustive analysis of the generated samples categorized by space group. Both successful and challenging instances are thoroughly scrutinized, contributing to a comprehensive understanding of the model’s capabilities.

We appreciate the reviewer’s positive feedback. Our aim was to provide a comprehensive demonstration of the model’s capabilities by performing thorough analyses of the model’s ability to generalize to structures very dissimilar from those seen in training.

Cons

The paper falls short in presenting convincing performance results. It lacks a demonstration of a substantial advantage over previous diffusion-based methods regarding the quality of generated structures and computational efficiency.

We regret that the reviewer is not convinced by the model’s performance results; however, we note that CrystaLLM does advance the state-of-the-art in the benchmarks presented, on both the CSP and unconditional generation tasks. Whether a result demonstrates “convincing” performance is a matter of opinion. Moreover, the performance results can also be limited by the nature and usefulness of the benchmarks themselves. For example, the best results on the unconditional generation structural validity

benchmarks are all already at 100%.

We disagree with the reviewer’s opinion that a demonstration of a substantial advantage over previous diffusion-based methods is lacking. We argue in the paper how the LLM approach (and, in particular, the CrystaLLM approach) comes with various advantages and flexibilities that diffusion-based methods do not have. In particular, the flexibility in terms of its inputs suggests that CrystaLLM may be conditioned on other properties of the structure, particularly those not traditionally included in the CIF format. Furthermore, as a large language model, it can leverage the established practice of fine-tuning, allowing the pre-trained model, and the rich internal representations it has captured, to be adapted for the prediction of materials properties. There is far less precedent in fine-tuning models based on diffusion and variational autoencoder architectures for tasks involving regression or classification.

There are a number of other distinct advantages over diffusion-based methods:

1. CrystaLLM supports both conditional and unconditional generation seamlessly, without requiring any architectural adjustments. The user simply provides more/less information in the prompt accordingly. On the other hand, DiffCSP requires architectural changes to support unconditional generation, and CDVAE also requires an architectural change to support conditional generation.
2. CrystaLLM natively supports space-group constrained generation, with no changes or external processing required. Conversely, the authors of DiffCSP devised an entirely different approach in DiffCSP++ to handle space-group constrained generation. They rely on a template retrieval and substitution method when the space group is unknown. In contrast, CrystaLLM generates a suitable space group automatically. There is no extra work required. The DiffCSP++ template-based approach consequently makes it impossible to propose structures when no suitable template exists, which is a limitation that CrystaLLM does not have. CDVAE does not support space group-constrained generation at all.
3. LLM-based models offer the ability to trade diversity for quality by adjusting the sampling temperature when generating structures unconditionally. This is a capability unique to this class of models.

Currently, the more recent diffusion-based models UniMat and MatterGen seem to be closed, with no publicly accessible source code repositories. This limitation means that comparisons must rely solely on the descriptions provided in their respective pre-prints. The exact number of parameters in each model and their training requirements in terms of GPU resources and time remain unclear. Based on the architectural details mentioned in the UniMat pre-print and the size of the original 3D U-Net model, we estimate that UniMat might have 800 million or more parameters (although this is a rough estimate and subject to considerable uncertainty). UniMat therefore appears to be less compute-efficient compared to the significantly smaller CrystaLLM model. Additionally, unlike CrystaLLM, UniMat does not support space-group constrained generation. MatterGen is a diffusion-based generative model distinguishable by its capability for fine-tuning through adapter modules. This feature makes it an exception amongst the diffusion-based models, which typically lack a precedent for significant fine-tuning. However, like the CDVAE and DiffCSP models, the adapter approach used by MatterGen represents an architectural adjustment that is required to be able to extend its capabilities. Moreover, the MatterGen study does not provide results for the traditional unconditional generation or CSP benchmarks, making it challenging to compare directly with models like CrystaLLM.

The differences above between CrystaLLM and competing diffusion-based methods indicate that CrystaLLM has the unique advantage of being a more flexible, general-purpose model, capable of supporting a number of different generation use cases, without requiring the user to switch between architectural variants, or different models entirely. CrystaLLM can alternate seamlessly between unconditional generation (when neither composition nor space group is known), generation conditioned on composition only, and generation conditioned on both composition and space group. In short, it has the potential to act as a foundation model for materials chemistry, as it automatically supports a wide range of generation use cases, and can be fine-tuned (in principle) for property prediction tasks. We have added a discussion of these points to make clearer the distinctive advantages of our model in section 2.3 of the revised manuscript.

Finally, we appreciate the reviewer’s point regarding computational efficiency. Comparing inference times across different models, such as in [2] and CDVAE, presents challenges due to variations in how these times are computed, and the fact that these models have different GPU requirements. For instance, [2] uses “5 batched generations and computes the average time to completion”. It is not clear to us how this method accounts for the variability in the number of tokens generated per CIF file, which can significantly impact inference times. Even so, we propose that the largest CrystaLLM model, with 200M parameters, will almost certainly perform more efficiently than the fine-tuned LLaMA 7B model in [2], given its radically smaller size, containing more than an order of magnitude fewer parameters.

Crucial evaluations, such as the validity and coverage of randomly generated samples, are noticeably absent.

Although CrystaLLM was not originally conceived as an unconditional generative model, we have nonetheless evaluated the model on a series of established benchmarks for unconditional generation, and now report results for the Perov-5, Carbon-24 and MP-20 benchmarks using metrics such as COV-R and COV-P. We found that CrystaLLM performs surprisingly well on these tasks, which highlights the flexibility of the approach. We therefore thank the reviewer for raising this issue and prompting us to test the model beyond its original intention. The performance on these benchmarks, comparison to other LLM models and diffusion models, and a discussion of the strengths and weaknesses of the different approaches is now in section 2.3 of the manuscript.

2. Lack of methodological comparison with related works. The idea of applying LLM to science domain generation has been explored in [1, 2]. While the concept of applying LLMs to science domain generation has been explored in prior research [1, 2], the paper fails to highlight fundamental distinctions between the proposed method and these existing approaches.

With our new extended comparison to other models in terms of unconditional generation, we have added discussion about the difference between methodologies in section 2.3. The models in [1] and [2] are similar to CrystaLLM in the sense that they are LLMs, but have important differences in terms of training data, tokenization, encoding and training (as noted by reviewer 2), in addition to crucial architectural differences, such as the number of parameters. CrystaLLM also pre-dates [2] and was submitted before [2] was published on arXiv. Indeed, [2] cites CrystaLLM. Nonetheless, we do extensive comparison to both of these papers (where reference results are available for them) and discuss differences. A more detailed discussion of the differences, strengths and weaknesses of the different approaches has been added to the manuscript in section 2.3.

Additionally, compared to diffusion-based methods, the invariance of the modelled data distribution remains unexplored.

We are not completely clear on what the reviewer means by the “invariance of the modelled data distribution”. If it refers to the symmetry invariances of the generated crystals, this has been accounted for in our validity tests. Our validity test ensures that the content in the generated file is a consistent CIF, which means checking factors such as the space group of the generated crystal and comparing this to the model declared space group, and checking the site multiplicities against the space group multiplicities. In these tests, we achieve almost 100% matching (Table 1).

In [2], there is a test for how well a model (an LLM) has learned translational invariances, which involves applying a transformation and calculating a perplexity score, which they refer to as the Increase in Perplexity under Transformation (IPT). The lower the score, the more invariant the model is to these transformations, in terms of the probability of the resulting structure under the model. Unfortunately, a direct comparison on this basis between our model and the model in [2] would not be meaningful, as each model is based on a different underlying syntax and vocabulary. Moreover, the question of whether a model is invariant to translations is particularly relevant in [2] because their approach represents all structures using the P1 space group, which lacks symmetry operators. The perplexity score calculated in that paper is useful for comparison of the models in the paper relative to one another, and shows an interesting tendency for greater respect of translational invariance in larger models, but it would not allow comparison between our model and the model in [2]. Even so, as the perplexity calculated depends on implementation details that are not provided in that paper, or in any code associated with it, we wouldn’t be certain we have reproduced the calculation as intended.

3. Some claims were not well supported. The abstract asserts that the proposed approach challenges conventional representations of crystals and demonstrates the potential of LLMs for learning effective ‘world models’ of crystal chemistry. However, there is an absence of discussion or results confirming the model’s similarity to ‘world models’ or showcasing any significant ‘world knowledge.’

While we strongly believe that CrystalLLM does learn an internalized world model (see our response to reviewer 2’s comment), we recognize that we have not performed experiments explicitly addressing this question. As such, we have removed the reference to ‘world models’ in our abstract. We still include a mention of world models in our introduction, but its purpose is to provide the motivation for our work. We do not claim in the manuscript that we have directly established that CrystalLLM has a world model of crystal chemistry. Proving or disproving this claim would require demonstrating connections between the model’s internal representations and its outputs, and is far beyond the scope of our work.

Given [2], the claim in the introduction, “Here, we report the first LLM specifically designed for crystal generation” was not accurate.

Our claim that we report the first LLM specifically designed for crystal generation is accurate. Our work pre-dates [2] and, in fact, our pre-print is cited by [2]. Nonetheless, we think that statements of precedence do not add to the scientific merit of any work, so we have removed this claim from the introduction.

[1] Flam-Shepherd, D. & Aspuru-Guzik, A. Language models can generate molecules, materials, and protein binding sites directly in three dimensions as XYZ, CIF, and PDB files. arXiv preprint arXiv:2305.05708 (2023).

[2] Gruver, Nate, et al. “Fine-Tuned Language Models Generate Stable Inorganic Materials as Text.” arXiv preprint arXiv:2402.04379 (2024).

(We include these reviewer-provided references for context.)

Reviewer 1

Authors have revised the manuscript point by point. I recommend that the present version be accepted for publication.

We thank the reviewer for the strong support for our revised manuscript.

Reviewer 2

This reviewer thanks the authors for their detailed response and modification to the paper. The updated paper contains more discussion over related works and extends the experimental results to unconditional generation. These modifications definitely improve the paper.

We thank the reviewer for appreciating these improvements of our manuscript.

While the non-conditional generation results are appreciated, as pointed out in the previous comments, this reviewer still feels some DFT-based validation is desired for a publication at Nature Communications.

We have followed this recommendation by performing the unconditional generation of 1,000 structures, and assessing the stability of the novel compounds among them, using DFT. The results are reported in new section 2.6 of the revised manuscript.

In summary, of the 1,000 unconditionally generated CIF files, 900 were valid. Amongst the valid, there were 102 structures which were novel with respect to the training dataset (in principle, novelty rates could be tuned using the temperature parameter for the unconditional generation, in trade off with how reasonable the generated structures are, so there is scope for optimization in further studies). For those, we performed DFT relaxation, and compared their energies with the convex hull. The mean E_{hull} of the 102 novel structures was 0.40 eV/atom. Notably, we found that 20 structures were within 0.1 eV/atom of the hull, including 3 with $E_{\text{hull}} = 0.00$ eV/atom. New figure 6 in the revised manuscript shows the 4 most stable of the novel compounds. In the Supplementary Information file (Table 9) we added the full list of the the 20 unconditionally generated novel structures with E_{hull} below 0.1 eV/atom. This demonstrates that the model can be used to find novel stable structures.

For the MCTS method, the benchmark was conducted over the 20 most problematic cases that were manually selected. Can the authors report performance improvement from the MCTS decoding method in a more general setting?

We have also checked that the MCTS decoding procedure is indeed capable of improving the quality of generated structures when guided by an estimator of formation energy. We took the generated compositions of the 102 novel compounds, and performed MCTS decoding using 1,000 iterations on each, using ALIGNN to provide feedback. After MCTS, the mean E_{hull} of the 102 novel structures was reduced to 0.34 eV/atom, and 22 were within 0.1 eV/atom of the hull. The extent of the improvement introduced by the MCTS approach can be expected to be directly related to the quality of the surrogate energy evaluator (ALIGNN in this case)—future improvements in ML-based energy evaluators will make the MCTS more effective for this task. A paragraph discussing this has been added in Section 2.6.

Reviewer 3

The additional experimental results do not demonstrate a clear advantage of the proposed method over existing LLM-based and diffusion-based approaches. The contribution of this paper remains unclear, both in terms of experimental performance and as a technical solution to current challenges. The previous concerns have not been adequately addressed.

We are sorry to read that Reviewer 3, in contrast with the other two reviewers, remained unconvinced

about the specific contributions of our work. Since our previous revision, we have made explicit what contributions our work makes in the context of existing literature, including the advantages over emerging diffusion-based approaches. As an illustration of the interest that our contributions generate, our arxiv preprint keeps accumulating citations (16 at the time of this submission), despite the work not being formally published yet.

Response to Reviewer 2

I appreciate the authors' efforts in adding DFT validation and additional experiments on the MCTS search algorithm. From the DFT validation results, the proposed model seems inferior to state-of-the-art models such as MatterGen. That said, this field is developing at a fast pace, and I appreciate the DFT results that put the model's performance in context.

We thank the reviewer for acknowledging our efforts in adding DFT validation and additional MCTS experiments.

We do not agree with the reviewer's point that CrystaLLM is apparently inferior to models such as MatterGen. Both models were trained on and evaluated against different datasets. In the case of CrystaLLM, novelty was assessed against a dataset with over two million compounds, whereas in the case of MatterGen, novelty was assessed against a different dataset with roughly half as many compounds. Since there isn't yet a standardized procedure for this task, it's impossible to be sure which model is superior based on the currently reported results alone. This point highlights the lack of standardized benchmarks in this emerging field of generative models for materials generation and design, which is a general issue. Such benchmarks will ideally also consider the nature and character of the generated compounds. Additionally, we reiterate that CrystaLLM does attain the best result in a number of categories on the crystal structure prediction and unconditional generation benchmarks. Therefore, CrystaLLM should not be excluded from being a "state-of-the-art model".

That being said, we concur with and appreciate the reviewer's comment regarding the addition of the DFT results. These types of results are indeed crucial for helping the community understand the advantages and limitations of this rapidly evolving class of models.

I believe the evaluation for the MCTS search algorithm can be further improved as 102 is still a relatively small sample size. Since ALIGNN is used for getting formation energy signals, I encourage the authors to conduct a larger scale experiment and evaluate more aspects of the MCTS algorithm – what is the error bar of the improvement brought by the search algorithm?

We understand the reviewer's concern that the sample size of 102 may be insufficient to establish a statistically meaningful result. However, we have confirmed, by performing a statistical test for significance, that the sample size is indeed sufficient to establish that the MCTS procedure reduces the above-hull DFT energies of the novel compounds. We explain our reasoning below in more detail.

First, it should be clarified that the MCTS procedure does not participate in the unconditional generation of novel compositions (see next point), but only in the prediction of more stable structures for given compositions. The criterion of stability used by the MCTS approach in our work is the ALIGNN energy. In this sense, MCTS definitely works as intended: after MCTS, the ALIGNN energy decreased (or remained constant) for all the compositions, with an average energy change of -153 ± 15 meV/atom.

However, the ALIGNN energy evaluator is only approximate, and does not always predict DFT energy differences accurately. Also, ALIGNN predicts energies for the as-generated, unrelaxed compounds, while the DFT energies are obtained for relaxed geometries. As a result, the average improvement of E_{hull} does not correspond with the improvement in ALIGNN energies. The MCTS-induced improvement of the average DFT energy is -56 ± 15 meV/atom, reducing the average E_{hull} from 0.40 to 0.34 eV/atom. The question from the reviewer can then be reformulated as whether, given the sample size, this DFT energy lowering is still significant, or merely a statistical fluctuation. In other words, is the improvement introduced by MCTS in terms of ALIGNN energies maintained (with statistical significance) after evaluating the energies with DFT?

To precisely answer this question, we performed a statistical test of significance. The null hypothesis is that the MCTS procedure does not bring any improvement in the average E_{hull} obtained by DFT. In that case, the DFT energy changes (from the original structures to those generated by MCTS) would be just randomly distributed with zero mean. We can calculate what the probability p would be of obtaining our DFT results under the null hypothesis conditions. For a paired t-test we obtained $t=-3.7$, which means that the probability of the DFT energy change observed being a statistical fluctuation in either direction (two-sided test) is $p=0.0003$. This is well below the threshold of $p=0.05$ typically accepted for

statistical significance. We have checked that a Wilcoxon signed-rank test, that accounts for deviations of the distribution of paired differences from normality, gives a similar result. Therefore, we can definitely reject the null hypothesis: the advantage introduced by MCTS does survive the transition from ALIGNN to DFT energies, despite the limitations of ALIGNN. Not only are the ALIGNN energies improved by the MCTS approach, but the DFT E_{hull} energies are as well.

In section 2.5 of the revised manuscript, we have provided a summary of these arguments, which are expanded in Supplementary Note 6 of the SI, giving details of the statistical test. We also provide a CSV file with the raw results for transparency. In the manuscript we also provide error bars for the average energy improvements brought by the MCTS procedure both for the ALIGNN and the DFT energies, as requested; these were calculated as the standard error of the means.

Does the search algorithm reduce the diversity/novelty of the generated samples?

As mentioned above, MCTS doesn't participate in the unconditional generation process, only in the generation of potentially more stable structures for given compositions. Because MCTS is driven by an energy evaluator, when working optimally it should lead to the groundstate structure of a given composition. Therefore, it can be expected to reduce the number of novel structures generated, as indeed it does.

To illustrate this point, we can use an example. Imagine that the unconditional generator had generated the formula SrTiO_3 , the standard autoregressive operation of CrystaLLM may not produce the ground-state tetragonal perovskite structure (or any of the reported metastable perovskite structures). However, if working correctly the MCTS procedure should drive the structure generation part of the task towards lower energy (from ALIGNN) structures and ideally to predict the tetragonal perovskite groundstate. In this case, the autoregressive generation could produce a new unique structure, but the MCTS procedure should not.

Because the MCTS procedure is not intended for use in scenarios where high levels of uniqueness are desired, we do not discuss this further in the manuscript. We feel that our inclusion of MCTS results in the "Generating Novel Materials" section may have led to some conflation of the crystal structure prediction and unconditional generation tasks. To avoid this confusion for the reader, we have moved the discussion of the MCTS results on the unconditionally generated structures to Section 2.5 where the MCTS procedure is introduced.

I in principle recommend paper acceptance.

We thank the reviewer for the positive recommendation of our paper, and for all the comments and suggestions received so far, which have led us to make a better case for our model and to test its limits.